# Unraveling Molecular and Genetic Insights into Neurodegenerative Diseases: Advances in Understanding Alzheimer’s, Parkinson’s, and Huntington’s Diseases and Amyotrophic Lateral Sclerosis

**DOI:** 10.3390/ijms241310809

**Published:** 2023-06-28

**Authors:** Alexandru Vlad Ciurea, Aurel George Mohan, Razvan-Adrian Covache-Busuioc, Horia-Petre Costin, Luca-Andrei Glavan, Antonio-Daniel Corlatescu, Vicentiu Mircea Saceleanu

**Affiliations:** 1Department of Neurosurgery, “Carol Davila” University of Medicine and Pharmacy, 020021 Bucharest, Romania; prof.avciurea@gmail.com (A.V.C.); razvan-adrian.covache-busuioc0720@stud.umfcd.ro (R.-A.C.-B.); horiacostin2001@yahoo.com (H.-P.C.); glavan.luca@gmail.com (L.-A.G.); antonio.corlatescu@gmail.com (A.-D.C.); 2Neurosurgery Department, Sanador Clinical Hospital, 010991 Bucharest, Romania; 3Department of Neurosurgery, Bihor County Emergency Clinical Hospital, 410167 Oradea, Romania; 4Department of Neurosurgery, Faculty of Medicine, Oradea University, 410610 Oradea, Romania; 5Neurosurgery Department, Sibiu County Emergency Hospital, 550245 Sibiu, Romania; vicentiu.saceleanu@gmail.com; 6Neurosurgery Department, “Lucian Blaga” University of Medicine, 550024 Sibiu, Romania

**Keywords:** molecular genetics, neurodegenerative disease, molecular pathology, epigenetics, gene expression, therapeutic targets, biomarkers, Alzheimer’s disease, Parkinson’s disease, Huntington’s disease, amyotrophic lateral sclerosis

## Abstract

Neurodegenerative diseases are, according to recent studies, one of the main causes of disability and death worldwide. Interest in molecular genetics has started to experience exponential growth thanks to numerous advancements in technology, shifts in the understanding of the disease as a phenomenon, and the change in the perspective regarding gene editing and the advantages of this action. The aim of this paper is to analyze the newest approaches in genetics and molecular sciences regarding four of the most important neurodegenerative disorders: Alzheimer’s disease, Parkinson’s disease, Huntington’s disease, and amyotrophic lateral sclerosis. We intend through this review to focus on the newest treatment, diagnosis, and predictions regarding this large group of diseases, in order to obtain a more accurate analysis and to identify the emerging signs that could lead to a better outcome in order to increase both the quality and the life span of the patient. Moreover, this review could provide evidence of future possible novel therapies that target the specific genes and that could be useful to be taken into consideration when the classical approaches fail to shed light.

## 1. Introduction

Neurodegenerative diseases are, according to recent studies, one of the main causes of disability and death worldwide. Interest in molecular genetics has started to experience exponential growth thanks to numerous advancements in technology, shifts in the understanding of the disease as a phenomenon, and the change in the perspective regarding gene editing and the advantages of this action. However, the concept of genes is not, as one might consider, a late 20th century notion. Aristotle predicted the existence of genes through postulating that the mother had her characteristics encoded inside the menstrual blood, while the father had his inside the semen, In addition, Hippocrates’ theory resembled what Charles Darwin later described as “pangenesis” [1]. However, two breakthroughs came a few centuries later. The first was when the Czech scientist Johann Gregor Mendel coined the terms “recessive, discrete and dominant factors” by observing his hybridization experiments performed on peas [2]. Later that century, Wilhelm von Waldeyer familiarized the scientific world with the term “Chromosomen”, derived from the work of Theodor Boveri, who coined the notion of “Chromatinelemente” [3]. The Nobel prize for Physiology or Medicine was awarded, in 1962, to Francis Crick and James D. Watson, for discovering the key to understanding not only molecular genetics, but also the fundament of life itself—the molecular structure of nucleic acids. This discovery was a huge milestone that led to understanding the base structure of life—DNA, RNA, and the creation of proteins; however, it also led to the comprehension of how a small misplacement of some nucleotide subunit can lead to a plethora of changes in the created proteins, further developing the disease.

Neurodegenerative diseases are represented by a group of disorders that are usually associated with protein deposits or misfoldings leading to chemical changes, loss of function, and apoptosis inside the neurons of the brain and spinal cord [4]. They are chronic and progressive, and, most importantly, there are many treatments that slow the development of the diseases and help manage the symptoms but do not cure the actual problem. Molecular genetics play a very important role in understanding the mechanics of the neurodegenerative disorders as it leads to identifying certain genes that are associated with this type of pathology and can also be a way of finding more efficient treatments [5]. In addition, molecular genetics has played an important role in identifying the specific proteins implicated in forming aggregates inside the cells that lead to the apparition of neurodegenerative disorders. The most common types of proteins that are implicated in forming these aggregates are amyloid-β, tau protein, α-synuclein, and prion protein [4,6]. The neurodegenerative disorders have different types of mechanisms behind each one of them, presenting a unique symptomatology. The most common neurodegenerative disorders are Alzheimer’s disease, Parkinson’s disease, Huntington’s disease, and amyotrophic lateral sclerosis, each one of which is associated with different genes, and where the common element is the formation of protein aggregates that lead to changing the physical and chemical properties of the nervous cell [4,7].

## 2. Alzheimer’s Disease (AD)

Alzheimer’s disease is the most common neurodegenerative disease that occurs in humans, representing 70% of the total dementia cases [7]. At first, Alois Alzheimer observed, in 1901, the interesting behavior of a 51-year-old woman who suffered from sleep disorders, memory loss, and progressive confusion. During the autopsy, Alois Alzheimer discovered the presence of neurofibrillary tangles and neuritic plaques, concluding that the disease was caused by the agglomeration of these structures. Neurofibrillary tangles—hyperphosphorylated tau proteins and neuritic plaques—aggregate beta amyloids [8]. A study conducted by Gatz et al. found that environmental factors can influence to some degree the chance of developing AD in humans that have the predisposing genes [9]. The perspective of Alzheimer’s disease was governed for more than 30 years by the amyloid cascade formation, which finished with the formation of the beta amyloids. However, newer studies tend to identify the AD amyloid cascade as a simplified view of the pathophysiology involved in the disease, emphasizing the glymphatic system, as well as the Lipoprotein receptor-related protein-1; RAGE [10].

### 2.1. Amyloid Precursor Protein

The genetic landscape of AD is dominated by mutations of, firstly, the amyloid precursor protein (APP); these mutations generate autosomal dominant alleles leading to the development of early onset AD [11]. APP can follow either the amyloidogenic or the non-amyloidogenic path, being cleaved by three different secretases: alpha, beta, or gamma [12]. Physiologically, the APP is cleaved by the alpha or gamma secretases [13] during the non-amyloidogenic path. During the amyloidogenic path, the APP is cleaved by the beta and gamma secretases, resulting in *Aβ38*, *Aβ40*, and *Aβ42* [14]. Subsequently, Aβ plaques are formed in a process that starts from Aβ monomers and ends with the development of amyloid plaques. Of the three beta amyloids previously mentioned, *Aβ42* is the least soluble [15].

Naturally, studies have shown that APP, which is coded on chromosome number 21, is directly correlated with a higher risk of developing early-onset AD in trisomy 21 individuals [16].

The relevance of amyloid pathogenicity is explainable by taking into consideration a study conducted in 2012 by Jonsson, T. et al. [17] on a population of 1795 Icelandic subjects. This study proved that the *A673T* (rs63750847) (or A2T) substitution, also known as the Icelandic mutation, decreases the risk of developing Alzheimer’s (and cognitive decline associated with aging) by lowering the overall aggregation and production of *Aβ40* and *Aβ42* by approximately 40% [17,18]. However, a study conducted in 2014 on a population of 2641 Chinese subjects showed that this gene does not explain the longevity of the old Chinese subjects who participated in this study because A2T was not included in all their genomes [19]. Another study performed in 2015 on 3487 Danish subjects showed that the *A673T* mutation is present in only one subject (0.014%) [20], in contrast with 0.43% in the Nordic population [17]. On the other hand, the mutation *A673V*, a mutation which manifests itself in a homozygous state, is a gene that is linked with early-onset AD [21]. However, this mutation leads to a distinctive manifestation of the disease in comparison to dominant genes that determine AD: familial-inherited AD usually develops amyloid deposits inside the striatum [22]. In comparison, this mutation tends to avoid the striatum in the incipient phase, focusing the amyloid deposits inside the cerebellum [21].

There are genes that protect against AD, and conversely there are genes that lead to early-onset AD (Osaka and Arctic mutations) or Cerebral Amyloid Angiopathy (Dutch and Italian mutations). For example, there are four mutations that happen on the *E693* position on the APP coding gene, exon 17: the Dutch mutation (*E693Q*), the Osaka mutation (*E693del*), the Italian mutation (*E693K*), and the Arctic mutation (E693G), all resulting in a change of the 22nd amino acid in Aβ, thus developing a peptide called E22. *E693Q* (rs63750579) mutation leads to a peptide called *E22Q*, and *E693Q* mouse models identify features shared with human Alzheimer’s brain pathology [23]. This mutation’s clinical phenotype is called hereditary cerebral hemorrhage with amyloidosis (HCHWA-D), and occurs in patients suffering from recurrent strokes and dementia. Aβ accumulates in the cerebral vessel walls because its overall production increases with this mutation, resulting in the formation of deposits in the meningo cortical vessels that lead to cerebral amyloid angiopathy [24,25]. *E693del* (dbSNP ID: NA) creates a mutant peptide known as *E22Δ*, which has been shown in a study conducted by Uddin, M.S., Tewari, D., Sharma, G. et al. to lead to increased endoplasmic reticulum stress by increasing the oligomerization of Aβ [26]. Further stress is created, as determined by an overexpressed chaperone *GRP78* and by the expression of the *GFAP* (glial fibrillary acidic protein), by *E22Δ*, thus correlating AD with the glymphatic system [27]. More properties of *E22Δ* were derived in a 2008 study that discovered that this mutant peptide creates fewer overall amyloid deposits; these deposits, however, are more resistant to proteolysis [28]. Recent studies performed by McKnelly et al. show the destabilizing and cytotoxic effect E22 peptides have on in vitro cell membranes by disturbing the phosphatidylcholine and phosphatidylserine constituting the cell membrane. This analysis showcases that these peptides tend to have a disturbing effect on the cell membrane proportional with the quantity of positive charges in the molecule. Thus, the peptides are, in order from least charged to most charged: *E22Δ* (two positive charges), *E22G*, *E22Q* (both have three positive charges), and *E22K* (four positive charges), meaning that the Italian mutation generates the most cytotoxic Aβ [29] (Table 1).

### 2.2. Presenilin1

Presenilin1, part of the γ-secretase, is encoded by *PSEN1*, which is located on chromosome *14q24.3* [34]. The change in the nucleotides of this gene is responsible for approximately 70 to 80% of manifestations of autosomal dominant Alzheimer’s disease [35,36]. Naturally, mutations to this gene will determine an unnatural development of Aβ and, in general, *PSEN1* mutations have a negative effect on the cleavage activity of the γ-secretase, resulting in an increased production in *Aβ42* rather than *Aβ40* [37]. However, it is important to note that the overall amplification of amyloid production might not be the right answer to explaining why a change in nucleotides in the presenilin gene determines AD. Besides the amyloid cascade theory, there is another theory that derives from mouse studies, which indicate that presenilin is truly important in the processes of memorizing, learning, and nervous cell survivability [38]. Mutations that can happen on exon 4 of the *PSEN1* gene include *A79V*, *M84V*, and *L85P*. *A79V* is an autosomal dominant mutation that determines an amplified proportion of *Aβ42* in comparison to *Aβ40*, by decreasing the latter’s production [39,40]. In a 2018 study, Koriath et al. discuss the fact that four alleles have been identified with a frequency of 0.00014% in the gnomAD database, deducing that this mutation has a low penetrance [41]. *M84V* is another autosomal dominant mutation that determines multiple types of atrophies—temporal and frontal lobe and also cerebellar and cortical atrophy [42]. It determines a greater *Aβ42* to *Aβ40* ratio, this time by increasing the overall quantity of *Aβ42* [43]. Moreover, mutation *M84T* has also been linked to Alzheimer’s disease. The *L85P* mutant determines a greater ratio of *Aβ42* to *Aβ40*, but is also shown to increase the production of *Aβ43* [44]. Autopsies of people who suffered from this mutation showed an aggregation of amyloid inside the basal ganglia and cortex, but there is not yet enough evidence to support theories regarding the pathophysiology of this mutation [45]. Up to this moment, only one mutation on the intron 4 of the *PSEN1* gene has been identified, namely *Int4del* (also referred to as L113_l114insT), regarding the deletion of a G nucleotide in the splice region of the *PSEN1* gene, just after the exon4, and therefore transcribing into three altered transcripts, one of which codes a protein with an extra threonine in addition to the *PSEN1* protein; as well as two smaller transcripts due to the apparition of premature stop codons [46]. Studies have shown that this mutant leads to an increase in the *Aβ42* to *Aβ40* ratio, by decreasing the quantity of *Aβ40* and *Aβ38* [47,48]. The *M139V* mutation happens on exon 5 of the presenilin 1 gene. Subjects with this mutation develop Alzheimer’s disease without many distinctive clinical and morphological features [49]. Regarding amyloid formation, this mutant will lead to increased *Aβ42* and *Aβ43*, and an overall decrease in *Aβ40*, *Aβ38*, and *Aβ37* [50]. Another mutation of the presenilin 1 gene is seen on exon 7 and is called *S212Y*. Neuropathologically, this mutation resembles the typical AD neurofibrillary tangles and neuritic plaques [51]. Liu et al. determined that in this mutation, production of Aβ42 is increased [47]. Another mutation of the presenilin 1 gene associated with early-onset and late-onset AD is known as *A434C*. This change in nucleotides leads to accumulation of plaques and amyloids inside the neocortex, accompanied by neurofibrillary tangles, and hippocampal and amygdala gliosis [52]. A 2022 study suggests that this mutant leads to a different conformation that results in a particular mode of interaction with the γ-secretase, creating a larger quantity of *Aβ42* than normal [53]. 

### 2.3. Presenilin2

*Presenilin2*, part of the γ-secretase, is encoded by *PSEN2*. The *PSEN2* gene is found on chromosome lq42.13 and has a total of 12 exons [53]. Mutations in presenilin genes usually lead, as stated before, to early-onset Alzheimer’s disease. However, not all changes in the nucleotides that compose the *PSEN2* gene determine AD. For example, a 2018 screening discovered a mutation in a Belgian subject, now known as mutation *K82fs* (situated on exon 5), who suffered from frontotemporal dementia. This mutation seems to determine a drop in presenilin 2 production in the hippocampal region and in the frontal cortex; further analysis of this pathology revealed Pick’s disease [54]. Two other studies performed in 2020 and 2021 revealed a new mutation on this gene: *c.*71C>A* [55], which happens on exon 13 3’UTR. The fact that this mutation is in the untranslated region is showcased by the fact that it lies where miR-183-5p attaches to the gene, inhibiting the suppressing activity of miR-183-5p [56]. Moreover, pathology associated with this mutation is relevant in the diagnosis of AD, for example, a greater ratio of Aβ42/Aβ40 and hippocampal atrophy [55,56]. Furthermore, *M239V* is a mutation, discovered in an 1995 study, that lies on exon 8, and linked with early-onset AD [57]. Autopsies performed on subjects carrying this substitution showcased beta amyloid aggregations and tau NFTs, the golden standard for diagnosing Alzheimer’s disease (Figure 1, Table 2).

Thus, correlating the anatomopathological features discovered during autopsies with the familial information and heritage of these patients determined the classification of this gene as a gene that is linked with early-onset Alzheimer’s disease [58].

### 2.4. Apolipoprotein E

However, early-onset Alzheimer’s disease accounts for approximately 1–2% of the total cases of AD. The majority of reported genetic cases of AD are caused by mutations of *APOE* (apolipoprotein E). Apolipoprotein E is a glycoprotein that consists of 299 amino acids, and is created inside the central nervous system by a great number of glial cells, including microglias, astrocytes, the cells of the choroid plexus, mural vascular cells, and neurons that undergo stress. *APOE* is abundantly expressed both peripherally and centrally; however, due to the BBB, it exists as separate pools. Therefore, it is crucial to understand what roles each pool may play in AD pathogenesis as well as therapeutic opportunities they present. Peripherally, *APOE* is produced primarily by the liver and plays an essential role in redistribution and metabolism of lipids such as triglycerides, cholesterol, cholesteryl esters, and phospholipids through lipoprotein particles [59]. *APOE* isoforms are associated with different lipoprotein particles in peripheral circulation; for instance, *APOE4* tends to preferentially associate itself with triglyceride-rich particles, while *APOE2* and *APOE3* prefer high density lipoproteins (HDLs). APOE-mediated cholesterol and lipid transport is critical for proper CNS formation and repair. *APOE3* shows a greater effect than *APOE4* at stimulating neurite outgrowth after injury, hence its prevalence among astrocytes. *APOE4* alters structural reorganization of neurons, decreases expression of key synaptic proteins, and inhibits glutamatergic signaling critical for neuronal plasticity and network maintenance. APOE’s effects vary by cell type; for instance, it is expressed by astrocytes, microglia, pericytes, and oligodendrocytes under various circumstances. Therefore, in order to gain a comprehensive understanding of its role within the brain it is crucial to examine its structure, lipidation status, and biochemical properties among different cell types that express it [60]. Research has demonstrated that *APOE*, a protein found in the brain, plays an integral role in Alzheimer’s disease by its interaction with amyloid-beta protein. *APOE* was discovered co-depositing with Aβ in amyloid plaques, thus contributing directly to AD risk. Knocking out *APOE* in amyloid model mice alters their Aβ plaque morphology significantly, indicating it plays a crucial role in fibrilization and amyloid deposition processes.

Studies of APOE’s effects on amyloid pathology have demonstrated isoform-dependent effects, with *APOE4* having the greatest impact, followed by *APOE3* and then *APOE2*. Studies indicate that those carrying an increased level Aβ compared to *APOE3* carriers and earlier deposition, with greater overall deposition, wider cortical distribution, and earlier deposition onset than its *APOE2* counterpart, while delaying deposition, less severe pathology, and protecting cognitive function, were seen with these carriers [61].

Studies have also demonstrated that *APOE4* stabilizes soluble, cytotoxic Aβ fragments and enhances fibrillogenesis to accelerate early amyloid pathology seeding. Thus, their interaction may serve as a potential target for therapeutic intervention at early stages of amyloid disease progression [62,63].

*APOE* plays an essential role in clearing away antibodies via several mechanisms, including receptor-mediated clearance and proteolytic degradation. Neurons utilize LRP1 receptors to absorb Aβ /APOE complexes from neurons via LRP1; this process is impaired for carriers of APOE4 due to reduced complex stability between *APOE4* and Aβ. Soluble Aβ can also be removed by proteolytic enzymes; however *APOE4* proves less effective at this than *APOE2* or *APOE3*, leading to reduced clearance overall [64].

### 2.5. APOE and Tau

One defining characteristic of Alzheimer’s disease (AD) pathology is the formation of neurofibrillary tangles (NFTs). NFTs consist of hyperphosphorylated tau aggregates as well as Aβ plaques. Studies have demonstrated that carrying the *APOE4* allele increases tau phosphorylation more than either *APOE2* or *APOE3*, particularly when exposed to Aβ oligomers. Furthermore, PET imaging studies on humans with this allele reveal greater tau deposition regardless of plaque presence. Additionally, neuronal *APOE4* was found to promote tau phosphorylation and cell death more effectively than *APOE3* in induced pluripotent stem cell cultures; animal models indicate that this genotype was also associated with higher total tau and phospho-tau levels, exacerbating tau-mediated neurodegeneration through modulating microglial activation [65]. Recently published research has demonstrated that deletion of Astrocytic *APOE4* can significantly decrease tau-related synaptic degeneration and disease-associated gene signatures, protecting against microglial phagocytosis as well as providing protection from tau. One study using AAV-tau delivery found that *APOE2* may cause tau phosphorylation and aggregation to increase, potentially due to formation of tau/APOE complexes primarily produced when non-lipidated *APOE2* was present. Recent genome-wide association study (GWAS) results indicate that *APOE2* may offer protection from AD risk by differentially regulating protein phosphatase 2A (PP2A), an important tau phosphatase in the human brain, unlike the detrimental impact of *APOE4* [66,67]. Taken together, these results show how the impact of *APOE* on tauopathy pathogenesis and tau-mediated neurotoxicity depends on which isoform is chosen. APOE’s role in tau pathology has drawn much interest both within AD research and among researchers studying related tau-related diseases such as FTD (frontotemporal dementia), CTE (chronic traumatic encephalopathy), and CBD (corticobasal degeneration). For instance, FTD patients carrying the *APOE4* genotype display earlier onset tau pathology, more severe neurodegeneration, and greater cognitive decline than non-APOE4 carriers, suggesting *APOE* may influence tau pathology independently of Aβ pathology. Therefore, understanding its molecular mechanisms within tauopathy may provide an important insight for developing strategies against AD and related neurodegenerative conditions such as FTD or CTE/CBD [68].

Studies conducted to date have demonstrated that *APOE* binds to regions of tau that contribute to pathogenic NFT formation, and one potential mechanism is that *APOE* may bind tau and block its phosphorylation sites. This interaction has been shown to be isoform-specific, with *APOE3* showing stronger binding affinity to tau’s microtubule-binding region than *APOE4*. According to research, reduced binding affinity of *APOE4* to tau may increase GSK3-mediated tau hyperphosphorylation and subsequent formation of NFTs. Alternately, some experts hypothesize that *APOE4* inhibits the Wnt signaling pathway through LRP5/6 receptors by increasing GSK3 activity and leading to tau phosphorylation. Current research is further exploring these potential mechanisms so as to understand APOE’s contributions in tau pathogenesis [69].

### 2.6. APOE and Neuroinflammation

The recent literature indicates that inflammation is an integral component of neurodegeneration, with its modulation by *APOE* gaining increasing attention. *APOE* may contribute to AD pathogenesis through various pathways; however, evidence is mounting that suggests they converge into neuroinflammation. Microglia cells often surround plaques found in postmortem brain tissue and play an active role in orchestrating an inflammatory response and clearing out amyloid plaques from memory cells. Studies conducted on mice lacking *APOE* have demonstrated decreased microglial response to plaques, suggesting it may be necessary for proper microglial activation in response to amyloid aggregation. Emerging research has also demonstrated that disease-associated microglia (DAM) or microglial neurodegenerative (MGnD) phenotypes exhibit a consistent transcriptional signature across Alzheimer’s mouse models, with APOE serving as a central regulator. *APOE*’s effect on microglial function appears to vary depending on its isoform, with recent research showing that APOE3 induces more effective microglial responses to Aβ injection than its isoform APOE4. This observation could be explained by Triggering Receptor Expressed on Myeloid Cells 2 (TREM2), which interacts with *APOE* with high affinity to modulate microglial responses. Evidence indicates that binding of *APOE* to TREM2 depends on both its isoform and lipidation status, potentially explaining differences in microglial function between isoforms. *APOE4* may impair homeostatic microglial functions due to reduced lipidation or affinity with TREM2, possibly accounting for its less potent homeostatic microglial responses compared with other isoforms [70,71]. Studies have demonstrated that C-reactive protein (CRP), produced by hepatocytes and released into plasma or serum, can be modulated by an individual’s *APOE* genotype in their peripheral immune system. CRP is an inflammatory protein produced in response to inflammation or injury and its levels vary accordingly. Proteomic analysis of cerebrospinal fluid has demonstrated lower CRP levels among *APOE4* carriers compared with individuals carrying either *APOE3* or *APOE2*. CSF samples also reveal reduced concentrations of CRP and complement cascade proteins among these carriers, in comparison with individuals who carry either *APOE2* or *APOE3* [72]. However, in spite of this tendency in the genotype, AD prevalence increases sharply with increasing serum CRP levels, with its greatest impact seen among *APOE4* carriers. However, in a longitudinal cohort, the *APOE* haplotype, but not the CRP haplotype, was associated with life-long cognitive decline, thus disproving any association between CRP and cognitive decline. Therefore, those carrying the *APOE4* allele may experience abnormal immune reactions, in response to pathological development, that lead to injury responses and cognitive deficits [73]. Therefore, targeting APOE-mediated inflammatory responses as part of therapeutic approaches for Alzheimer’s disease or neurodegeneration could prove useful and should be explored further as a potential solution.

### 2.7. Important APOE Mutations Involved in AD Onset

#### 2.7.1. *c.-488C>A*

The biological impact of this variant found within the *APOE* promoter remains unknown [30]. It falls within the functional domain of the HuD protein that spans nucleotides −651 to −366 [31] which has been shown to act as a negative regulator in multiple cell types including PC12 neuronal-like rat chromaffin cells, SK-N-SH neuroblastoma cells, C6 glial cells, and U373 astrocytoma cells. Furthermore, substitution of nucleotide 488C will remove potential binding sites used by transcription factor EGR1, which would interact directly with this transcription factor [32].

#### 2.7.2. *c.-24+38G>A*

Yee et al. conducted a study that sequenced the *APOE* genes of 257 Southern Chinese individuals spanning 69 AD patients, 83 subjects with mild cognitive impairment (MCI), and 105 cognitively healthy controls in South China. Two AD patients (1.4%), one control (0.5%), and no MCI patients (0%) carried this variant; it was detected globally at an incidence frequency of 0.00033 in the gnomAD variant database, with most carriers having East Asian heritage (0.0037 frequency; 43 heterozygotes) [32].

#### 2.7.3. *c.-24+288G>A*

Yee et al. conducted a study where this variant was identified in 257 Southern Chinese individuals spanning Alzheimer’s disease (AD), mild cognitive impairment (MCI), and cognitively healthy controls from Southern China. One AD patient (0.7%), one MCI patient (0.6%), and one control (0.5%) carried it [32]; gnomAD reported the variant *c.-24+288G>A* as having a worldwide frequency of 0.00016 with only five heterozygote carriers of East Asian origin worldwide. Conversely it was significantly more prevalent among East Asians with an East Asian ancestry population, with an incidence rate of 0.015 according to gnomAD v2.1.1 (Oct 2022); out of 22 carriers listed there, all were had an East Asian ancestry with at least one homozygote from this region.

#### 2.7.4. *c.-23-377A>G*

This variant was identified in a study which involved sequencing the APOE genes of 257 individuals of Southern Chinese origin, comprising 69 AD patients, 83 subjects with MCI, and 105 cognitively healthy individuals—including six AD patients (4.3%), three MCI patients (1.8%), and three controls (1.4%) [32].

In the gnomAD variant database, this variant was reported at an overall frequency of 0.00073 and at A much higher frequency among individuals of East Asian ancestry; 22.2 carriers were identified from East Asia alone with one being homozygous for it.

#### 2.7.5. *A18T*

Yee et al. conducted an in-depth analysis of 257 Southern Chinese individuals’ APOE genes, comprising AD 69 patients, 83 subjects WITH mild cognitive impairment, and 105 healthy controls, and found one AD patient (0.7%), three MCI patients (18%), and four controls (1.9%) [32].

Zhou et al. identified this variant as one of six *APOE* variants with potential clinical relevance and functional consequences due to its high prevalence in at least one population and predicted deleterious effects by in silico algorithms. They performed whole genome and exome sequencing analyses from 138,632 individuals from different populations and discovered that this variation alters the *APOE* signal peptide sequence at its cleavage site, potentially hindering secretory efficiency by disrupting recognition at this spot [33] (Table 3).

### 2.8. Microtubule-Associated Protein Tau

The discovery that microtubule-associated protein tau (MAPT) gene mutations caused frontotemporal dementia with parkinsonism linked to chromosome 17 (FTLD-17) was a historic moment, providing genetic evidence of dysfunction within tau alone as sufficient cause of neurodegeneration independent of Aβ. Abnormal accumulation of tau is seen across various central nervous system disorders such as AD, Pick’s disease (PiD), progressive supranuclear palsy (PSP), corticobasal degeneration (CBD), and argyrophilic grain disease; thus, targeting tau offers the possibility not only of treating AD itself but also of treating many other tauopathies associated with Aβ. Human brains produce six distinct isoforms of tau through alternative splicing of the *MAPT* gene located on chromosome 17q21. The different isoforms result from alternatively splicing exons 2 and 3, leading to variants with zero (0N), one (1N), or two (2N) N-terminus inserts [74]. Exon 10 can also affect protein production, leading to three (3R) or four (4R) microtubule-binding domains residing on C-terminal tau proteins based on whether they contain three (3R) or four (4R). 3R tau binds less tightly than 4R tau, so that six tau isoforms could exist: *3R0N*, *3R1N*, *3R2N*, and *4R0N* are all possible isoforms. An average brain contains equal levels of 3R and 4R tau. However, in certain tauopathies—for instance, those linked to frontotemporal dementia with parkinsonism linked to mutations near exon 10 on chromosome 17 (*FTDP-17*)—there may be an increase in 4R tau, increasing interaction with microtubules [75,76]. Tau undergoes various post-translational modifications during both normal physiological processes and stress-induced responses, such as glycosylation, ubiquitination, glycation, nitration, and oxidation processes, with phosphorylation being the most widely studied. When exposed to healthy brain environments such as Alzheimer’s disease or other tauopathies such as multiple myeloma or parkinsonism, the levels of tau phosphorylation vary; in healthy brain tissue there are around two or three residues, while neurodegenerative conditions such as Alzheimer’s or tauopathies involve much higher phosphorylation, with nine or ten phosphates per molecule being created by imbalanced activity between tau kinases and phosphatases. This results in hyperphosphorylated tau being localized within its environment, in turn resulting in multiple serine/threonine/tyrosine residues on different places on its protein structure due to an imbalance between its kinase/phosphatase activity [77]. Glycosylation, ubiquitination, glycation, nitration, and oxidation are among the many post-translational modifications that play a role in controlling tau during both normal and stress-induced responses. Of these modifications, phosphorylation has been widely studied. Tau is typically phosphorylated on two to three residues in healthy brains. However, in AD and other tauopathies, its hyperphosphorylation occurs at nine phosphates per molecule. This imbalance results from disruptions in the activity of tau kinases and tau phosphatases, leading to decreased affinity of tau for microtubules as well as resistance against degradation by both ubiquitin-proteasome pathway degradation and calcium-activated neutral proteases. Hyperphosphorylated tau forms fibrils and aggregates into NFTs over time. Major tau kinases include GSK-3b, CDK5, PKA, and MAPK CaMK II MARK [78,79], while PP2A has been identified as the primary dephosphorylation enzyme for abnormal tau. Changes in tau kinases and phosphatases have long been documented as markers of AD and related conditions, with expression and activation rates of tau kinases and phosphatases often increasing over time [80]. A variety of processes, including Aβ, impaired brain glucose metabolism, inflammation, and infection all play a part in abnormal tau hyperphosphorylation; therefore, identifying pathways governing post-translational modifications of tau may prove extremely valuable when searching for therapeutic targets.

Research has demonstrated that an abnormal hyperphosphorylation of tau occurs prior to its accumulation in Alzheimer’s disease-affected neurons. This hyperphosphorylated tau has been identified both within neurofibrillary tangles as well as within the cytosols of AD brains. Utilizing mAβ Tau-1 for immunocytochemical studies has demonstrated that abnormally phosphorylated tau (not normal tau) accumulates in neurons without tangles (stage “0” tangles) in Alzheimer’s and aged hippocampi [81]. At present, tau found in neurofibrillary tangles is known to be ubiquitinated while abnormally hyperphosphorylated tau isolated from AD brain cytosol does not display this property, indicating abnormal hyperphosphorylation occurs prior to its accumulation into neurofibrillary tangles. Davies et al. [82] demonstrated that tau phosphorylation occurs prior to PHF formation in the AD brain by employing monoclonal antibodies targeting mitotic phosphor epitopes. One possible explanation for abnormal hyperphosphorylation of tau is conformational changes occurring within diseased brains that make it an even more favorable substrate for phosphorylation and/or dephosphorylation, respectively. Moreover, they have developed monoclonal antibodies to detect conformational changes in tau, and have demonstrated that tau indeed undergoes conformational changes both in AD patients and transgenic mice that overexpress human tau [82]. As in *FTDP-17*, which is caused by certain missense mutations of tau, these mutations make tau more susceptible to hyperphosphorylation by brain protein kinases and lead to its hyperphosphorylation. However, in AD it is less likely that tau mutations alone are responsible for hyperphosphorylation; several neuronal proteins become over-phosphorylated due to an imbalance between protein phosphorylation and dephosphorylation processes. Biochemical analyses have demonstrated that Alzheimer’s brain tissue contains excessive levels of tubulin and neurofilaments that have become hyperphosphorylated; immunocytochemical analysis shows neurofilaments and *MAP1B* to also be affected. Furthermore, both PHF-abnormally hyperphosphorylated tau and its cytosolic counterpart are readily dephosphorylated by in vitro phosphatases [83].

### 2.9. Important MAPT Mutations Involved in AD Onset

#### 2.9.1. *MAPT IVS10+12 C>T*

This mutation was identified as the causative mutation in the Kumamoto pedigree, a Japanese kindred with frontotemporal dementia [84]. Primary clinical symptoms included parkinsonism and dementia manifesting during their fifth decade, with an average onset age of 53 years and length of illness lasting seven years (n = 6).

Brain tissue from affected individuals was observed to contain elevated exon 10 tau transcripts and 4R tau isoforms, with elevated exon 10 tau aggregates observed both in neurons and glial cells; isolated tau filaments displayed twisted ribbon-like morphologies made up of hyperphosphorylated 4R tau. Yasuda et al. reported neuropathological findings for one member of this pedigree while Takamatsu et al. discussed neuropathological findings for one individual within this pedigree [84,85].

#### 2.9.2. *MAPT A152T*

In a large series of American and European people, the *A152T* variant was found to be associated with an increased risk of DLB, but not PD [86]. The variant was found in 10 out of 2456 controls (minor allele frequency 0.20 percent). Among PD patients, 18 out of 3229 carried the variant (MAF 0.28 percent), and among DLB patients, six out of 442 patients carried the variant (MAF 0.68 percent). In addition, two out of 181 patients with multiple system atrophy carried the variant (MAF 0.55 percent), a non-significant increase in frequency.

Consistent with the variable clinical presentations associated with this variant, neuropathological reports are similarly diverse. Abnormal tau accumulation appears to be the unifying feature in all cases for which postmortem findings are available. In some cases, prominent Lewy body pathology is seen [87]. In other cases, the pathology is indicative of PNLA, as indicated by the prominent neuronal loss and tau deposition in the globus pallidus, subthalamic nucleus, and substantia nigra, with lower levels of pathology in the motor cortex, striatum, pontine nuclei, and cerebellum.

This variant has been shown to impair tau’s ability to bind microtubules, resulting in less efficient microtubule assembly and impaired microtubule stability. In addition, although the mutant protein appears to aggregate with lower efficiency than wild-type protein overall, it is more prone to oligomer formation [88]. Isogenic human iPSCs generated from fibroblasts of an *A152T* carrier showed that the mutant tau is predisposed to proteolysis by caspases and other proteases and leads to greater tau pathology.

#### 2.9.3. *MAPT K257T*

Autopsy analysis revealed Pick’s disease, a subtype of FTD characterized by severe frontotemporal atrophy, particularly in the temporal lobes. The neocortex, hippocampus, and some subcortical regions displayed numerous tau-positive Pick bodies while diffuse hyperphosphorylated tau was detected in certain cell bodies [89]. Recombinant tau protein with the *K257T* mutation displayed reduced capacity to facilitate microtubule assembly [89].

#### 2.9.4. *MAPT L266V*

Kobayashi et al. reported one case who underwent autopsy that revealed severe frontotemporal atrophy with Pick-like pathology, evident by prominent atrophy of both frontal and temporal cortices as well as caudate nucleus and substantia nigra, prominent neuronal threads, coiled bodies, and ballooned neurons throughout all layers of the cortex and the brainstem; abundant tau-positive inclusions within neurons and astrocytes with a high concentration of tau-positive inclusions were present throughout all cortical layers and the brainstem; there were abundant tau-positive inclusions found among neurons and astrocytes, with a high concentration found throughout all cortical layers as well as a high concentration found within the caudate nucleus; and numerous neuronal threads, coiled bodies, and ballooned neurons were observed. Additionally, neuronal threads and ballooned neurons were observed—all were present and notable for its severity [90].

Hogg et al. reported another case characterized by severe frontotemporal atrophy with Pick-like pathology, as well as significant atrophy of the hippocampus and parietal lobe. Neuronal loss was extreme across the cortex and substantia nigra with severe atrophy in these regions; there was also significant gliosis present, while tau-positive inclusions were widely distributed, including within the hippocampus, striatum, and substantia nigra. Pick bodies were Gallyas silver positive and contained straight filaments distributed randomly throughout layers of cortex [91].

In vitro, this mutation alters exon 10’s splicing, leading to higher levels of tau transcripts, with four microtubule binding repeat domains (4R tau). This leads to decreased rates of microtubule assembly induced by tau and lower tubulin polymerization levels; more specifically, with 3R tau isoforms being more likely to assemble than their 4R counterparts (Table 4).

### 2.10. The Evolving Landscape of Alzheimer’s Disease Donanemab Treatment: Exploring Current and Future Perspectives

Donanemab, a humanized antibody that targets the N truncated pyroglutamate-amyloid-b peptide (*pGlu3Aβ, AβpE3*), has shown potential to reduce cerebral amyloid depositions in Alzheimer’s disease, constituting a promising treatment option for AD. This therapy aims either to reduce *pGlu3* Aβ formation at glutamyl cyclase, or to clear *pGluAβ* after formation and/or block aggregation [92,93,94]. Donanemab was found to be highly active against amyloid, especially cored plaques within the CNS. However, its efficacy as a treatment of AD is still uncertain. The binding properties of antibodies targeting *AβpE3* are different against the soluble and aggregated forms of *AβpE3-42* [95,96].

Lowe et. al. [97] proved that Donanemab shows good tolerance to dosages up to 10 mg/kg, with a terminal half-life mean of four days following a single dose of 0.1–3 mg/kg. The half-life was increased to 10 days with a dosage of 10 mg/kg. A 40–50% decrease in amyloid levels was seen at 24 weeks with a Standardized Uptake Ratio (SUVR), which decreased from 1.65 to 0.36, and a change in Centiloids (CLs), which decreased from −44.4 (SD 14.2) from baseline. Moreover, 90% of the subjects developed antidrug antibodies 3 months after a single dose.

In a separate report by Lowe et al. [97], it was shown that Donanemab caused rapid amyloid decreases even after just one dose [98]. The mean reduction in PET amyloid was −16.5 CL at 10 mg/kg, −40.0 CL at 20 mg/kg, and −50.6% at 40 mg/kg. The multiple-dosage groups at week 24 showed a mean reduction in amyloid levels in the 10 mg/kg Q4weekly arm, a 50.2 CL for the 10 mg/kg Q2weekly arm, and a 58.4 CL for the 20 mg/kg Q4weekly arm. In both the single-dose and multi-dose cohorts, some patients had an amyloid clearance level below 24.2. Donanemab was effective in treating all but one patient (97.8%) [98].

In another cohort, nearly 40% of participants (46 of 115) receiving Donanemab reached the full amyloid clearing threshold (24.1 CL), and their baseline amyloid levels were lower than those of the group as a whole. In the first 24-week period, there was a moderately negative correlation (r = −0.54) between the baseline amyloid level and that of the plaques removed. The amyloid clearing was sustained, with a very low rate of re-accumulation (0.02 mean rate over one year). Participants who had an amyloid concentration of =11 CL by week 24 but discontinued treatment, would require about 3.9 to accumulate amyloid up to 24.1 CL. The overall tau accumulation was reduced by 34% in the Donanemab groups compared to placebos at week 76 [99].

In another study by Lowe et al., two patients experienced asymptomatic amyloid imaging abnormalities as a result of cerebral microhemorrhages. One patient discontinued treatment because of these reactions. In a second study by the same authors, seven serious adverse effects were reported in six patients. Only one patient died from a non-drug-related myocardial ischemia. In another study by the same authors [98], seven serious adverse events were reported among six patients, with only one patient dying due to a non-drug-related myocardial infarction. Two patients (4%) from the interventional arms stopped taking Donanemab because of adverse reactions.

Another investigation relates that the rate of mortality from all causes was lower for participants who received Donanemab (0.76%) compared to those who received placebo (1.6%). The study also found that there were no differences between the groups in terms of the rates of serious adverse reactions, which were respectively 19.85% and 20.0% in the Donanemab group and the placebo group [99].

As a conclusion, Donanemab is now being tested as an alternative therapy for Alzheimer’s. It has been a long time since the urgent need to slow the disease’s progression was met. Alzheimer’s disease treatments have been approved by the FDA, but there has been controversy over this approval. There is also a need for better and more effective treatments. According to a systematic review of phase III trials for preclinical Alzheimer’s, the first anti-amyloid therapies were performed. It is important to plan carefully, to perform longitudinal assessments, and to store and manage data effectively as clinical trials and new therapeutics are developed. The research conducted will determine Donanemab’s effectiveness for a diverse population, increasing the retention of the treatment and improving referrals to clinical trials.

### 2.11. Brief Reflection Point

The section offers an engaging panorama of Alzheimer’s research progress and challenges, and advocates for an interdisciplinary global collaboration to overcome them. With each step we take towards unravelling Alzheimer’s complex web, the promise of effective interventions—perhaps eventually leading to cures—becomes more tangible, bearing immense significance for millions across the globe who suffer.

## 3. Parkinson’s Disease (PD)

Parkinson’s disease (PD) is a progressive neurodegenerative condition, most often seen among elderly individuals worldwide. It is estimated to affect between 0.3% of the general population and 1–3% of those over the age of 65. By 2030, its numbers are expected to climb from 8.7 million to 9.3 million. James Parkinson first described PD symptoms in 1817, and they typically include dysfunctions of the somatomotor system, including rigidity, bradykinesia, postural instability, gait dysfunction, and tremors.

Disease progression leads to progressive degeneration of the nigrostriatal dopaminergic pathway, leading to significant neuron loss in substantia nigra pars compacta (SNpc) neurons and depletion of dopamine (DA). Non-motor dysfunctions such as dementia, hyposmia, and gastrointestinal abnormalities often accompany disease progression.

Pathological hallmarks of Parkinson disease (PD) include accumulations of a-synuclein aggregates known as Lewy bodies or neurites in certain areas of the central nervous system, such as the basal ganglia, dorsal motor nucleus of vagus (DMV), olfactory bulb (OB), locus coeruleus (LC), intermediolateral nucleus in spinal cord (IML), celiac ganglia, and enteric nervous system (ENS) [100].

New research indicates that Parkinson’s disease (PD) neuropathology could be caused by environmental stressors and the natural process of aging itself. Exposure to environmental toxins, drugs of abuse, or the stress of aging may lead to chronic low-level inflammation in the brain, leading to something known as “inflammageing,” and thus to neuron cellular senescence.

Pathologically, Parkinson’s patients typically display damage in the substantia nigra pars compacta and pontine locus coeruleus regions of their brains characterized by depigmentation, neuronal loss, and gliosis. By the time symptoms manifest themselves, approximately 60–70% of neurons from this region have already been lost [101].

Genetic factors have been estimated to account for roughly 25% of the risk associated with Parkinson’s disease, and genetic variants associated with it vary both in terms of frequency and risk. While rare mutations within individual genes (known as monogenic causes) may contribute to its development (known as monogenic causes), these were generally discovered through linkage analysis in affected families using linkage analysis; some common genetic variants that only contribute a small amount to risk were also discovered via genome-wide association studies (*GWASs*), including many common genetic variants that contribute an intermediate risk, such as *GBA* or *LRRK2* variants.

Genetic classification of Parkinson’s can lead to various treatment approaches and prognoses for each subgroup, often depending on age of onset, family history, and pathogenic variant presence; age at onset, family history and presence of pathogenic variants are frequently used as criteria for stratifying this form of PD. Monogenic forms may or may not represent typical forms of idiopathic PD. Importantly, some genes involved in monogenic PD have also been identified through *GWAS* studies as common variants. One such gene, *SNCA*, which was discovered through these analyses to have common variants, is also implicated in monogenic PD pathogenesis, supporting the role of a-synuclein. Other pathways may also play a part in its pathogenesis, such as tau aggregation, which is linked with other neurodegenerative conditions such as Alzheimer’s and frontotemporal dementia [102].

Familial Parkinson’s, also referred to as Mendelian or monogenic PD, is characterized by rare yet high-penetrance genetic variants that increase risk. Autosomal dominant (e.g., *SNCAA53T* and *VPS35D620N*) and recessive forms of familial Parkinson disease have been identified using linkage analysis in families with the help of next-generation sequencing technologies, though only 5–10% of cases fall under these single gene variants. Conversely, low-penetrance genetic variants with more frequent associations with sporadic Parkinson’s disease have been identified through genome-wide association studies (*GWASs*). At first glance, distinguishing familial from sporadic disease may help with diagnosis, prognosis, and genetic counseling for at-risk family members; however, such classification may obscure shared genetic or biological mechanisms that underlie them both.

An example is that both rare and common genetic variants associated with *SNCA* have been shown to increase Parkinson’s risk, underscoring its role as an aSyn-mediated disease mechanism. Missense variants in *SNCA* such as *p.A53T*, *p.A30P* and *p.E46K* cause autosomal dominant familial Parkinson disease, while the common risk variant *SNCArs356168* occurs in approximately 40% of European-ancestry populations and has only modest effects on disease risk [103].

*SNCA*, or synuclein complex A, is a 14.5 kDa protein consisting of 140 amino acids encoded by 5 exons and having a transcript length of 3041bps. Located on *4q21.3-q22* of human chromosome 4, this synuclein protein family also includes *SNCB* (*5q35*) and *SNCG* (*10q23.2-q23.3*). The structure of *SNCA* protein comprises an N-terminal region with incomplete KXKEGV motifs, an extremely hydrophobic NAC domain, and an acidic C-terminal domain. Under physiological conditions, it appears as either an intrinsically disordered monomer or helically folded tetramer structure. Although it was previously thought to be toxic in this form, recent observations have refuted this idea.

Over the last two decades, various hypotheses have been put forward regarding the toxic structural form of *SNCA*; none has yet been unanimously agreed upon. What is known is that its neurotoxic form accumulates within neurons before disseminating throughout anatomically interconnected regions in the Parkinson’s disease brain through interneural transmission using various mechanisms.

Although *SNCA* is most abundantly expressed in the brain, it also appears in heart, skeletal, muscle, and pancreas cells. While its exact function remains undetermined, several hypotheses have been proposed based on its structure, physical properties, and interactions with interacting partners. *SNCA* may play an essential role in regulating dopamine release and transport, inducing microtubule-associated protein tau fibrillization and exerting a neuroprotective phenotype in non-dopaminergic neurons by modulating both p53 expression and transactivation of proapoptotic genes leading to decreased caspase-3 activation.

Given *SNCA*’s central role in neurodegenerative processes, its essentiality may suggest that selective forces among sarcopterygians play a vital role in modulating its molecular and cellular mechanisms. Fine-tuning of these mechanisms through minute changes to protein activity could have contributed to evolutionary adaptations that meet different environmental and ecological needs. Current evidence indicates that amino acids 32 to 58 of *SNCA*’s N-terminal lipid binding domain are critical to its normal cellular functioning and disease pathogenesis. Lineage-specific substitutions could have led to structural remodeling and functional adaptation in *SNCA* over generations, and any mutation affecting its critical regions is likely to be harmful. These discoveries provide the framework for investigating their critical roles through various interaction studies as well as targeting them with drug discovery efforts to treat FPD [104].

Mutations in *LRRK2* account for 5–12% of familial parkinsonism cases and 1–5% of sporadic cases. So far, seven missense *LRRK2* mutations have been identified as pathogenic: *R1441G*, *R1441C*, and *R1441H* were all found to be pathogenic; these variants can be found within different functional domains of *LRRK2*, including *R1441G* located on *R1441C*, which affects *R1441H*; *Y1699C* was also involved, as well *G2019S*, *R1628P*, *G2385R*, and *I2020T* variants specific to certain populations. The *G2019S* mutation, which leads to constitutive activation of the kinase, is one of the most prevalent. It accounts for an estimated 36% of familial and sporadic Parkinson’s cases among North African Arabs; approximately 30% among Ashkenazi Jewish populations; up to 6% among familial cases in Europe and North America; and up to 3% among apparently sporadic cases; however, it does not occur among Asian populations. Various other *LRRK2* mutations such as *G2385R, R1628P, S1647T, R1398H*, and *N551K* have also been associated with parkinsonism within certain Asian populations. Studies conducted among Asian populations spanning Singapore, Taiwan, and mainland China have established that *LRRK2* variants *G2385R* or *R1628P* may increase risk for Parkinson disease. Furthermore, the *G2385R* variant has been found to increase risk for Parkinson’s disease among Japanese and Korean populations; these variants were not seen among Indians and Caucasians. Although *LRRK2* mutations exist in familial PD, no differences exist in clinical features or neurochemical differentiation between idiopathic and familial forms of parkinsonism. Both forms of Parkinson disease (PD) involve profound dopaminergic neuronal degeneration and gliosis in the SNpc, decreased dopamine levels in the caudate putamen, and Lewy body pathology in the brainstem; therefore, understanding *LRRK2* plays an essential role for all forms of PD [105].

Mutations in the *PINK1* gene are an important cause of early-onset Parkinson’s disease (EOPD), accounting for 1–9% of genetic cases and 15% of early-onset cases—second only to Parkin mutations. First identified by Unoki and Nakamura in 2001, its 18 Kb span contains 8 exonic regions that encode for an essential serine/threonine protein kinase essential for mitochondrial functioning and metabolism.

As reported by the MDSGene database, worldwide there have been 151 *PINK1* mutation carriers who carry 62 different disease-causing sequence variants involved with both sporadic and familial Parkinson disease cases; 13 definitely pathogenic mutations exist alongside 44 possibly pathogenic variants (13 definitely pathogenic mutations and 44 possibly pathogenic variants).

*PINK1*, an encoded protein from the *PINK1* gene, primarily localizes to mitochondria where it serves as a serine/threonine-type protein kinase that regulates mitochondrial quality control (mitoQC). MitoQC involves maintaining respiring mitochondrial networks while selectively eliminating damaged ones through mitophagy, an essential process critical for cell homeostasis. Furthermore, in addition to mitoQC functions, *PINK1* also plays an anti-death, pro-survival role under various forms of stress conditions, preventing neuronal cell death under various stress conditions. Additionally, its protein contains an N-terminal mitochondrial targeting sequence (MTS or TMD), transmembrane sequence (TMS or TMD), and C-terminal domain [106].

### 3.1. Perspectives of Treatment

The metal-based hypothesis of neurodegeneration is an attractive explanation for the pathophysiology behind Parkinson’s disease. This hypothesis proposes that reactive oxygen species are generated by redox-active metals, particularly iron. ROS (reactive oxygen species) cause membrane phospholipids to be peroxidized, resulting in the production of reactive aldehydes. Both ROS and reactive aldehydes modify α-synuclein, causing it to aggregate. Aggregated α-synuclein causes mitochondrial dysfunction, resulting in a vicious cycle of increased ROS production and decreased ATP synthesis. In order to provide a more effective treatment of PD, a multi-task strategy targeting these events is needed [107].

Coenzyme Q10 is a vital antioxidant that is important in reducing oxidative stresses, a factor implicated in Parkinson’s disease and other neurodegenerative diseases. In order to establish their potential as a marker of disease, a number of studies have investigated the levels of CoQ10 found in different tissues of people with PD or other parkinsonian disorders. Several studies have also explored the therapeutic potential of CoQ10 for the treatment of PD or PS. Several clinical studies have examined the ability of ubiquinol (the antioxidant form of Coenzyme 10 or CoQ10) to reduce oxidative damages observed in PD. CoQ10 can restore mitochondrial function by bypassing Complex I dysfunction, which is a feature of sporadic PD. A meta-analysis consisting of 8 controlled trials involving 899 patients found that CoQ10 is well-tolerated, safe, and does not improve motor symptoms compared to placebo. The study authors do not recommend CoQ10 as a routine treatment for PD except in cases where levodopa is wearing off [108].

Recent clinical trials have shown that iron chelation therapy is a promising approach to treating Parkinson’s disease. Due to the multifactorial nature of PD, targeting a specific factor, such as iron, may not be enough for complete neuroprotection. It may be necessary to develop and test multifunctional drugs that combine the iron chelation process with other protective properties.

A growing global population is aging, and central nervous system disorders such as Parkinson’s and Alzheimer’s diseases are becoming more prevalent. These disorders are linked to iron accumulation in certain areas of the mind. Finding effective treatments for these conditions is therefore crucial to improving the longevity and quality of life of elderly people [109]. DFO (deferrioxamine) was administered intramuscularly in early studies of Alzheimer’s patients. DFP (deferiprone) was the first oral chelator used to treat Friedreich’s Ataxia [110]. This condition is characterized by frataxin deficiency, which is the mitochondrial chaperone for iron. Animal studies have shown that DFO or DFP can reduce iron in different brain regions, and also provide neuroprotection for an animal model of Parkinson’s disease. In two clinical trials, oral DFP was administered to PD patients in cohorts. MRI measurements showed that the iron content of the substantia nigra, as measured by DFP, decreased. UPDRS scores also improved. Iron chelation was not effective in patients who had high levels of inflammatory marker IL-6. DFP has a major problem with agranulocytosis, and neutropenia. This requires testing of white blood counts every week and complicates logistics. DFP is currently being tested in phase II clinical trials on early-stage PD [107,111].

α-Synuclein is another point of interest regarding the treatment of PD. Although clinical trials using monoclonal antibodies to treat α-synuclein aggregates have failed to show any improvement in Parkinson’s symptoms, other studies in progress or recruiting participants may prove that targeting α-synucleinopathies as a therapeutic option is possible.

### 3.2. Brief Reflection Point

This section leaves us with a clear message: while we have made significant strides toward understanding Parkinson’s disease, much remains unsaid. To truly make progress against it possible for millions worldwide living with Parkinson’s, interdisciplinary research must continue as we unravel its complexities. Continuing the search for knowledge is not simply essential scientifically; it also enhances quality of life for millions affected by this condition worldwide.

## 4. Huntington’s Disease (HD)

Huntington’s disease represents a neurodegenerative disorder that bears the name of the American physiologist George Huntington (1850–1916). He was the first one to observe the main manifestation of this disease, characterized by uncontrolled motor activities that tend to be compared to dance-like movements of the body, named chorea, as well as abnormalities regarding both the personality and patient’s way of thinking. The motor disturbance can be split into two different stages: the incipient stage, characterized by a hyperkinetic syndrome, due to the loss of the medium spiny neurons (MSNs) of the indirect pathway, followed in the later stages of the disease by the loss of the MSNs of the direct pathway, which lead to a hypokinetic syndrome [112]. Moreover, Huntington was the one to observe that this pathology has both a hereditary nature as well as a progressive onset, saying “Once it begins it clings to the bitter end” [113]

However, Huntington’s disease represents a rare pathology, with an incidence of 10.6 to 13.7 out of 100,000 individuals. The statistics vary across ethnic groups due to the differences in the HTT gene. According to a study conducted by Bates et al. in 2015 [114], the average length of repeated CAG trinucleotide sequences varies from 18.4–18.7 in the European population and 17.5–17.7 in the East Asian population.

As it was increasingly studied, it was shown that Huntington’s disease represents an autosomal dominant progressive neurodegenerative disorder, consisting of a repetitive set of (CAG)_n_ trinucleotide sequences in a gene found on the chromosome 4p16.3, between *D4S10* and *D4S98* [115] (Gusella et al., 1983), called huntingtin (HTT), leading to a polyglutamine expansion. This repetitive sequence of trinucleotides leads to a mutation of the huntingtin gene (mHTT).

However, the number of CAG units repeated in an allele has a strong significance when it comes to predicting whether the allele is a disease generating one or not. The normal range for the healthy population is between 6 and 35 units. Between 36 and 39 units, the disease is not guaranteed to occur, but there are also chances of developing it. Over 40 units, the mutation is regarded as highly penetrant and it will generate a phenotype along the adult population [116].

Conversely, the length of the CAG repeated sequence is not only correlated with the chances of developing the Huntington’s disease, but also with the onset age of this pathology. The study conducted by Persichetti et al. (1994) showed that the bigger the sequence of trinucleotides, the earlier the beginning of the neurological symptoms [117].

Additionally, other variations have been identified in the HTT gene beyond its polymorphic/expanded CAG repeat. These include modifications in both its coding sequence (such as an expanded CCG repeat after CAG repeat and deletion polymorphism at codon 2642), as well as untranslated sequences, intron sequences, and those flanking its centromeric and telomeric ends. These variations have been used to define HTT haplotypes, which represent groups of sequence variants on specific chromosomes that tend to remain relatively unchanged between generations due to limited recombination events in this relatively small segment of genome [118].

These haplotypes, carrying expanded alleles in HD patients, have revealed that approximately 50% of Europeans with HD have one common ancestor, while multiple independent mutations on different chromosomal backbones account for the rest. However, none of the most frequent haplotypes found either on HD chromosomes or among HD heterozygotes appear to significantly alter motor diagnosis age. Therefore, while natural sequence variation at HTT might occasionally serve as a source of disease modification in HD, its contribution is not significant enough [119].

The mHTT gene leads to the formation of an abnormal huntingtin protein with an extended polyglutamine tail at the NH2-terminal end.

The toxicity of the mHTT gene is generated through the formation of two kinds of mRNA. The first is represented by the HTT mRNA, while the second is a HTT mRNA exon 1, which encodes only the first exon due to the CAG repeated sequence [120]. Therefore, from the first type of mRNA will result a full-length huntingtin protein made of huntingtin fragments, which are loops that are used for proteolytic cleavage and will later represent the site for complex post-translational modifications that will lead to the HTT exon 1 fragment, as well as some individual boxes. The HTT mRNA exon 1 on the other side translates itself with the help of the ribosome just in a huntingtin fragment, which is arranged in three parts. The first is described as a mixed sequence of 17 amino acids, the second part is the polyglutamine sequence, also called polyQ, while the last part of the exon 1 fragment is characterized by a proline-rich domain (PRD) (Figure 2).

Upon translation, either the full-length huntingtin protein or the HTT exon1 protein is generated. The HTT exon1 fragment comprises the HTTNT sequence, the polyglutamine sequence encoded by the CAG repeat, and a proline-rich domain. On the other hand, the full-length huntingtin protein includes the HTT exon1 sequence, as well as ordered and disordered protein segments represented by boxes and loops, respectively.

Proteolytic cleavage, which occurs at recognition sequences located in the disordered segments, leads to the formation of various products, including HTT exon1-like fragments. Fragments with expanded polyQ segments play an important role in the development of Huntington’s disease through molecular mechanisms that have yet to be fully understood.

Huntington’s disease begins early, before any symptoms have emerged, with transcriptional dysregulation taking place due to mutations of HTT that disrupt transcriptional machinery via interactions with transcription factors and molecular mediators such as CBP (cAMP response element-binding protein).

Recent research has demonstrated changes to chromatin remodeling through impaired histone activity, and reductions in mitogen-activated and stress-activated protein kinase 1 (*MSK-1*) activity among striatal neurons of Huntington’s disease patients and animal models. Overexpression of *MSK-1* resulted in increased expression of peroxisome proliferator-activated receptor gamma coactivator alpha (PGC-1a), a transcriptional co-activator involved in mitochondrial biogenesis that may also protect against neuronal death [121].

Histone deacetylase (HDAC) inhibitors have shown promising results in animal models of Huntington’s disease, and may represent a viable therapeutic target. Class III HDACs (sirtuins) have demonstrated promise as neuroprotective targets, with one study revealing an improvement in motor function and reduced brain atrophy in a Huntington’s disease mouse model by overexpressing *Sirt1*, an NAD-dependent protein deacetylase. Notably, *Sirt1* also restored brain-derived neurotrophic factor (BDNF) [122].

The following actions that happen after the translation are represented by a condensation and oligomerization of the protein fragments in cytoplasm, which will lead to a dysfunctional proteostasis of the cell. Moreover, these fragments will go through an aggregation process inside the nucleus of the cell, binding to the DNA as inclusions and therefore altering the entire process of transcription. All of these pathological processes will alter both the axonal transport and inter-synaptic transmission, as well as the mitochondria of the cell, causing a decreased energy output [121].

According to a study conducted by G. Vonsatellel et al. in 1998 [123], the mutated huntingtin protein can be found in both the dystrophic neurites and in nuclear inclusions of the neuron, with a higher prevalence in the cortex and neostriatum in comparison with the globus pallidus and cerebellum, where this type of mutated protein cannot be found. Moreover, this study attested that mHTT protein was found in 38% to 52% of the neurons of the patients with juvenile HD (with an age under 20 years) and in 3% to 6% of the neurons regarding the adult onset of the HD.

A study conducted by the GWAS (Genome-Wide Association Study) in 2017 discovered the existence of a correlation between the HD onset and a gene called *MSH3*, that together with the *MSH2* gene, will lead to a heterodimer called MutSβ, whose main goal is to repair the possible mismatches of the DNA after replication. However, a variant of *MSH3* seems to be involved in the somatic expansion of the CAG repetitive sequence, leading to an increased risk of developing HD, affecting mostly the brain striatum, which is the most affected by HD. This test was conducted using the post hoc analysis [124].

However, even if today we understand the pathogeny of HD better than at any previous time, the therapeutic directions are quite limited, and it is difficult to achieve a conclusive result. According to a study conducted by Travessa et al. [125], of 99 trials that proposed to study 41 compounds to treat HD, only 2 trials made it to phase 4 (2%), with a success rate of only 3.5%.

### 4.1. Treatment

Tetrabenazine is the only drug approved to treat Huntington’s chorea in North America and some European countries; however, this could increase depression as a potential side effect. Post hoc analysis showed that advanced Huntington’s disease patients who already took antidepressant drugs did not experience worsening depression after starting treatment with tetrabenazine. Antipsychotic drugs are most often employed for chorea treatment in Europe, while both tetrabenazine and antipsychotics are applied at an equal level in North America and Australia.

Amantadine’s efficacy for chorea treatment remains inconclusive, while pridopidine, a dopaminergic stabilizer, has been assessed as a symptomatic therapy in Huntington’s disease; trials showed mild stimulatory and inhibitory effects depending on dopaminergic tone levels.

However, a recent phase III trial involving 437 patients failed to demonstrate significant improvements in modified motor score after six months of treatment with doses up to 90 mg/day; further analysis suggested potential benefits in UHDRS total motor score as an endpoint. Huntington’s disease offers few treatment options for cognitive dysfunction and behavioral abnormalities, and trials of cholinesterase inhibitors have proven unsuccessful in improving cognitive dysfunction. Moreover, addressing the behavioral abnormalities remains challenging.

Therapies designed to lower levels of mutant huntingtin (*mHTT*) represent one of the most promising approaches for disease modification. These emerging therapies target either DNA or RNA of the *mHTT* gene; targeting can take the form of antisense oligonucleotides (ASOs), RNA interference (RNAi), or small molecule splicing inhibitors. Currently, ASOs are being investigated in a human phase 1b/2a study delivered intrathecally that catalyze degradation of *HTT* mRNA via RNAse H; in animal models, this approach resulted in a decrease of up to 80% in *HTT* mRNA levels over time [126].

Attaining lower levels of *mHTT* may prove particularly effective for Huntington’s disease modification. One approach aimed at this goal involves targeting either DNA or RNA of the *mHTT* gene; targeting its expression using ASOs, RNAi, or small molecule splicing inhibitors are possible solutions.

RNAi-based approaches use RNA molecules that bind to mRNA in the cytoplasm and prompt its removal by Argonaute 2, an RNAse enzyme found within an RNA-induced silencing complex. While therapeutic strategies using this approach are still in their preclinical stage, one treatment could potentially provide permanent *HTT* reduction through intracranial injection into the striatum of a small molecule splicing modifier that has shown promising results in animal models with muscular atrophy; screening is underway to identify small molecule modulators of *mHTT*.

Targeting the DNA of *mHTT* can be done using two approaches: zinc finger proteins and the CRISPR/Cas9 system. Zinc finger proteins are structural motifs that bind directly to DNA; synthetic zinc finger transcription factors targeting CAG have been used successfully to lower levels of *mHTT* protein in animal models. However, as they create non-native proteins, they could potentially trigger immune reactions, and further research is required before conclusively targeting them.

CRISPR/Cas9 is a groundbreaking gene editing technology, allowing scientists to make precise changes to an organism’s DNA. CRISPR stands for Clustered Regularly Interspaced Short Palindromic Repeats found in bacteria and archaea genomes; the repeats are connected by “Cas” genes which encode for proteins such as the Cas9 enzyme.

CRISPR/Cas9 works by employing a small RNA molecule called a guide RNA (gRNA), designed to target specific DNA sequences with high specificity. Engineered specifically to bind to its target sequence, gRNA allows for highly targeted binding between the DNA molecule and itself—with Cas9 enzyme acting like “molecular scissors” at this location, cutting its DNA at target locations. More recently it has also become a genome editing tool with various applications in human disease treatment. Huntington’s disease patients were treated using this technology to excise promoter regions, the transcription start site, and CAG mutation expansion of the *mHTT* gene found in their fibroblasts, leading to permanent allele-specific inactivation of this gene. This approach has also been successfully tested in an HD rodent model, providing proof of concept. Further preclinical work needs to be completed before CRISPR/Cas9 gene editing technology can reach clinical application. Recently raised concerns over unexpected off-target mutations have necessitated more study [127].

No treatment has yet been shown to stop or slow Huntington’s disease from progressing, although various clinical trials have investigated its potential effectiveness, including of minocycline, riluzole, and remacemide. However, none has shown significant effects.

Coenzyme Q10 (CoQ10) at 600 mg/day was observed to slow functional decline, although this did not reach statistical significance. Additional research is evaluating effects of doses up to 2400 mg/day as well as testing on preHD gene carriers. Safety and tolerability data showed that doses of up to 3600 mg/day were well-tolerated by most study participants without experiencing serious adverse events [128].

Creatine is widely acknowledged for its antioxidant properties and potential to improve mitochondrial function and cell bioenergetics. Unfortunately, an earlier study conducted using 10 g/day did not demonstrate significant improvements in total motor score, functional capacity, or neuropsychological testing scores compared with controls. Therefore, phase III studies are now taking place with higher dosages up to 40 g/day as part of CREST-E (Creatine Safety Tolerability Efficacy in Huntington’s disease) [129].

### 4.2. Brief Reflection Point

In essence, this section reaffirms the significance of continued and interdisciplinary research in unraveling Huntington’s disease. As our understanding deepens, so does the potential to enhance diagnosis, develop effective treatments, and ultimately improve the lives of those affected by this disease. Our journey in decoding Huntington’s is far from over; each step forward holds promise and hope for millions globally.

## 5. Amyotrophic Lateral Sclerosis (ALS)

Amyotrophic lateral sclerosis is a neurodegenerative disease usually manifesting with an onset of focal muscular weakness and fatigue, having the tendency to spread selectively among upper and lower motor neurons; in some cases dysarthria, dysphonia, and dysphagia can also occur [130]. It affects the health state of the patient as it spreads typically from distal muscles to more proximal ones. The survival rate is about 2 to 5 years from the date of diagnosis, with the cause of death usually being respiratory failure. ALS can be classified as being familial (10% of the cases) and sporadic (90% of the cases), both of which show similarities between symptomatology. This shows how important epigenetic studies are in discovering new treatments for ND and how both epigenetics and environmental factors can determine the apparition of ALS and neurodegenerative disorders in general [131]. Various studies have shown that ALS has a slightly higher incidence in the female population. Furthermore, great geographical discrepancies show that there is a prevalence in the European region, while in Asia and the Middle East, far fewer ALS cases have been reported [132,133,134]. In addition, it has been proven that environmental exposure to different agents such as infectious agents or heavy metals can lead to genetic alterations and can also facilitate the development of degenerative diseases. The most common genetic genes that cause ALS are *C9orf72, SOD1, TARDBP, FUS,* and *TBK1* [130,131] (Figure 3).

Numerous studies have confirmed to date that *C9orf72* mutations are the most common in familial ALS disease, which only represents 10% of the ALS cases ([135]), compared to other gene mutations that lead to the apparition of this pathology. *C9orf72* is a gene that is formed out of 11 exons and it is implicated in splicing mechanisms that produce transcripts and 2 isoforms. The mutation that is widely spread throughout ALS cases especially in Europe and North America is the GGGGCC hexanucleotide expansion, located in the first intron of variants 1 and 3 and in the promoter region of variant 2 [132,134]. More than 22 repeats were found in neurodegenerative disorders and it has been stated that the patients that present the GGGGCC hexanucleotide repeat might share the Finnish founder risk haplotype; this statement that can be supported by the fact that numerous studies have shown that the expansion is most common in the Finnish population compared to other regions [136,137]. The mechanisms behind this expansion can be explained by its capacity to be transcribed into repetitive RNA, which then forms both sense and antisense RNA foci and five dipeptide repeat proteins (DPRs). This can explain possible mechanisms of pathology represented by dysfunction of *C9orf72* protein or toxic accumulations of either RNA foci, which can sequester RNA binding proteins, creating inclusions or DPRs inside the nucleus of the nervous cells [135,138].

Mutations of the *SOD1* gene represent one of the most common factors of apparition for familial ALS and represent about 10–20% of FALS (familial ALS) cases, being far less common in sporadic ALS cases. At the moment, there are over 155 gene mutations of the *SOD1* that can have implications in ND diseases, including ALS. These can result in different modifications to the *SOD1* protein regarding its level, structure, or enzymatic activity, which can eventually lead to toxic protein accumulations due to misfolding of the *SOD1* proteins [139]. It was first stated that the *SOD1* mutations are implicated in the apparition of ALS by a mechanism based on the loss of function of this protein, but this hypothesis was soon abandoned after some experimental studies [139,140,141]. The *SOD1* protein, also known as superoxide dismutase 1, is a very well-conserved gene located inside the nucleus, cytoplasm, and mitochondria’s membrane. It is formed by 5 exons that encode 153 amino acid metalloenzyme, binding Cu and Zn ions, forming a dismutase that removes radicals from the cells and metabolizes them into oxygen and hydrogen peroxide ([139]). The most common mutations of SOD1 associated with ALS are usually based on changes in the structure of the protein, specifically, the amino acid positions, and might include *A4V* [142], *G93A* [143], and *L84F* [144,145].

A4V mutation changes the alanine from codon 4 to valine in exon 1 and it is associated with aggressive forms of ALS representing 50% of the SOD1 mutation cases reported in North America [139,146]. *G93A* is a heavily studied gene mutation of the *SOD1* protein that represents a substitution of glycine with alanine from the codon 93 of the *SOD1* protein, changing its conformation, and is also responsible for approximately 20% of the familial ALS cases. According to a study conducted by the Chinese Pharmaceutical Association in 2023 [147], oxidative stress has a major impact on ALS pathology; patients showed different oxidative markers, such as glutamate excitotoxicity, and dysfunctions at several levels, such as mitochondria, due to calcium influx, and axon as well as protein oxidation. These modifications observed were at *SOD1-G93A* in mice. This mutation is relatively rare in the general population but it is very common in familial ALS, and multiple studies on animal models have also shown that having the *SOD1-G93A* mutation is enough to cause motor-neuron degeneration [147,148,149]. Understanding the mechanisms as well as the specific effects of *SOD1* mutations on protein structure and functions is an important area of research for developing effective treatments for ALS and neurodegenerative diseases in general.

The *L84F* mutation changes the amino acid at position 84 from leucine to phenylalanine, and is associated with a slightly mild form of ALS compared to other mutations such as *G93A*, resulting in both protein instability and misfolding that can lead to forming protein accumulations [144]. According to a study on the population of central Italy [145], the *L84F* mutation is a rare mutation that has been identified in several families as a result of a common ancestor who carried the mutation. In addition, the article reported several pathological characteristics of the patients with ALS who had this mutation, including young age of onset, a relatively slow progression of the disease, and the predominance of upper limbs’ motor neuron involvement, suggesting that the clinical features might be caused by the specific mutation (Table 5).

Mutations of the TARDBP gene are associated with neurodegenerative diseases, especially with amyotrophic lateral sclerosis; this gene encodes the *TDP-43* protein, which is implicated in the apparition of this ND. The exact mechanisms of how this mutation leads to motor neuron degeneration are not fully understood, but one of the reasons might be that it affects the structure of the function of *TDP-43*, thus leading to the formation of toxic protein aggregates [150,151]. According to an article published in *Nature Genetics* [152] that investigated the frequency of *TARDBP* mutations by sampling 93 familial ALS cases and 109 sporadic ALS cases, the mutations that were identified were mostly missense mutations, with *A382T* being the most common one found. In addition, regarding the clinical symptoms, the patients found with *TARDBP* mutations had an early onset age and usually bulbar onset, as well as a shorter survival rate compared to that of other ALS cases. The *TARDBP* gene was present in 5% of the familial ALS cases and 1.9% of the sporadic cases. TARDBP mutations interfere with the normal function of *TDP-43*, leading to abnormal protein clumps in motor neurons, which contribute to death and degeneration of these motor neurons. While its exact mechanisms for inducing ALS development remain enigmatic, speculation points toward changes in RNA splicing, protein translation, or degradation processes being involved. *TARDBP* mutations are an important genetic risk factor for ALS and studying them can provide insights into its mechanisms of development, potentially providing potential therapies targeted specifically towards these mutations [153]. *TARDBP* gene mutations have long been linked with amyotrophic lateral sclerosis. One of the more often-seen *TARDBP* mutations linked to ALS is *M337V*; this mutation can be found in both familial and sporadic cases of the condition. Other *TARDBP* variants that have also been frequently connected with this form of neurological disease include *A315T, G348C*, and *A382T* mutations [150,154,155].

The TDP-43 *M337V* mutation has been associated with familial amyotrophic lateral sclerosis (ALS) in Japan. A study conducted on 41 families living with familial ALS revealed that 11 of those families contained *TDP-43 M337V* mutations. The *TDP-43 M337V* mutation results in the production of an abnormal *TDP-43* protein that aggregates abnormally and accumulates in motor neurons, ultimately leading to their degeneration and death, and thus contributing to ALS development. It is thought that this accumulation may play a key role in its manifestation [154,155].

The *TDP-43 A315T* mutation has been linked with familial motor neuron disease. This genetic alteration occurs within the *TARDBP* gene, which contains instructions for producing the *TDP-43* protein; mutations disrupting this production can lead to abnormal protein accumulation within motor neurons and eventually death. According to [155], attaining abnormal levels of *TDP-43* through mutations such as *A315T* has been linked with degeneration and death of motor neurons, eventually leading to motor neuron diseases such as ALS. Although this mutation only appears in a small percentage of familial motor neuron disease cases, it is a pathogenic one responsible for inducing symptoms in those who carry it. Undertaking research into the *TDP-43 A315T* mutation and its impact on motor neuron disease could provide key insights into its underlying mechanisms and possible therapeutic targets, and genetic testing may play a pivotal role in identifying individuals carrying this mutation who could be at an increased risk of familial motor neuron disease. Another study shows that the *TDP-43* (A315T) transgenic mouse model provides an opportunity to investigate how mutations of the TDP-43 gene *A315T* affect amyotrophic lateral sclerosis and other motor neuron diseases, leading to abnormal accumulations of protein aggregates, which eventually cause degeneration and death of motor neurons. It reveals that transgenic mice carrying the *A315T* mutation of *TDP-43* may succumb to early death due to digestive complications before fully manifesting neurological signs associated with ALS, suggesting it also influences their digestive systems and may contribute to their early demise. Although the exact mechanisms underlying gastrointestinal complications remain poorly understood, experts speculate that abnormal *TDP-43* protein build-up in intestinal cells may lead to dysfunction and damage. The early death of these transgenic mice highlights the necessity of conducting further studies into the effects of *A315T* mutation on non-neuronal systems within the body, and possible therapeutic approaches for managing them. The *TDP-43* (A315T) transgenic mouse model provides an essential way of studying its underlying mechanisms as well as devising potential treatments against it [156].

The *TARDBP* gene mutation, *A382T*, has been associated with an increased susceptibility for motor neuron diseases such as amyotrophic lateral sclerosis. Studies have demonstrated that this mutation results in the production of an abnormal *TDP-43* protein that accumulates and damages motor neurons, ultimately leading to their decline and eventual demise. The *A382T* mutation is considered an uncommon pathogenic mutation found only in a small proportion of familial ALS cases. Further research is necessary to fully understand how this mutation leads to disease development and potential therapeutic approaches that target it. Genetic testing could prove invaluable in identifying individuals carrying this variant who may be at greater risk of ALS and related motor neuron diseases [157,158] (Table 6).

The Fused in Sarcoma (FUS) gene represents another common gene that can cause ALS apparition and encodes a protein involved in controlling RNA processing and transport within cells. Mutations of this gene have been linked with neurodegenerative diseases. Mutations that alter FUS protein accumulation in neurons may result in their dysfunction and apoptosis. Both missense and truncating mutations of FUS may alter protein function and disease development. Some mutations are inherited in an autosomal dominant manner, while others may occur spontaneously. Further research must be conducted in order to understand how FUS mutations work and to develop therapeutic approaches; genetic testing could help identify individuals at increased risk of FUS-related diseases [159,160].

Regarding amyotrophic lateral sclerosis (ALS), researchers have discovered several mutations of the FUS gene that may contribute to its progression. *R521C, R521H*, and *P525L* mutations are among the most frequently encountered, occurring in approximately 5–10% of familial cases and a smaller proportion of sporadic cases of ALS.

Mutations result in the production of a mutated FUS protein that accumulates in motor neuron cytoplasm, eventually leading to their degeneration and death. While the exact mechanisms underlying ALS development due to such mutations remain enigmatic, experts speculate that they disrupt RNA metabolism and transport processes that are vitally important in maintaining proper motor nerve functionality [159].

A previous study investigated the prevalence and clinical features of FUS mutations among Italian patients with familial ALS. From 54 cases studied, 2 individuals (3.7%) carried *FUS R521C* mutations that showed early disease onset, as well as more severe disease progression, than patients without this mutation, suggesting FUS mutations may play an integral role in familial ALS prevalence among Italian populations. This suggests FUS mutations may play a vital part in creating familial ALS within Italy’s population [161,162].

The *TBK1* gene is also implicated in ALS apparition and plays an essential role in regulating various cellular processes, such as autophagy, inflammation, and immune response. Mutations have been identified as one cause of ALS and FTD; such mutations result in abnormal protein aggregate accumulations within motor neurons, causing degeneration and death. Interference with regular cellular processes may play a part in contributing to this condition and to its progression. Mutations in *TBK1* have been identified as an uncommon cause of ALS and FTD. These mutations cause abnormal protein aggregates to build up in motor neurons, eventually leading to their degeneration and death, possibly interfering with normal cellular processes that contribute to disease development in this way, contributing further to ALS or FTD development [163,164]. Mutations in the *TBK1* gene have been identified in patients suffering from amyotrophic lateral sclerosis (ALS), including missense, frameshift, and truncating mutations. Some of the more prevalent mutations include the frameshift mutation *TBK1* p.Ile383Thr, which results in truncated protein expression [164]. Missense mutations such as *p.Arg357Ser* and *p.Gly290Val* have also been found; these latter two also appear in familial cases as these affect highly conserved amino acids found within its protein. Additionally, missense mutations impair its ability to bind its target proteins, allowing greater protein expression from within.

These mutations are thought to disrupt *TBK1*’s role in regulating various cellular processes, such as autophagy and immune responses, potentially contributing to motor neuron degeneration in ALS patients [164,165].

According to [164], various studies have demonstrated *TBK1* gene mutations among familial and sporadic cases of ALS, suggesting *TBK1* dysfunction may play an integral part in its progression. Furthermore, this piece delves into potential pathways through which these mutations could contribute to ALS progression, such as disrupted autophagy, neuroinflammation, and mitochondrial malfunction.

This article highlights the various functions of *TBK1* in terms of its involvement with innate immunity, autophagy, protein aggregate clearance, and neuroinflammation; these processes have been implicated as being causal in ALS. Studies have identified mutations of the *TBK1* gene among ALS patients, indicating its involvement. Furthermore, dysfunction caused by mutations may contribute to neuroinflammation, which results in protein aggregate accumulation as well as motor neuron death, resulting in the progression of the condition.

This article suggests that *TBK1* could be an effective therapeutic target for treating ALS, specifically by increasing autophagy and decreasing neuroinflammation. Furthermore, further research needs to be conducted in order to fully comprehend how this gene contributes to ALS, as well as to create effective treatments that target it [164,165,166].

In recent years, epigenetic studies have led to major discoveries regarding neurodegenerative diseases and appear to be the key element in finding a treatment that can cure the condition rather than just alleviate it. The term epigenetic refers to any genetic alteration that occurs at any molecular level, associated with several factors such as environmental agents or stress, that does not implicate the modification of the DNA directly [167]. The most common mechanisms regarding epigenetics are DNA methylation, miRNAs, and PTM (post-translational modification) of histones [168].

ALS is a disease characterized by the loss of neuromuscular junctions, and is also associated with apoptosis of UMN (upper motor neuron) and LMN (lower motor neuron) cells, as well as the surrounding astrocytes and microglial cells. It is also associated with inclusions in which the main protein involved is *TDP-43*, which can cause protein aggregation by its mislocalization between the nucleus and the cytoplasm [169]. In addition, there are multiple pathological causes of ALS; the most relevant are the failure of protein degradation, changes in RNA metabolism, and axonal transport.

The failure of protein degradation can be the leading cause implicated in the formation of protein aggregates that can disturb the homeostasis of the cells, and it is also one of the similarities between neurodegenerative diseases. Protein chaperones can aid the cells to the point of saturation by refolding the misfolded protein. This can explain the formation of protein aggregates; however, if the chaperones are overloaded, ubiquitin inclusions are formed inside the nucleus. The genes that lead to protein accumulation relating to ALS pathologies are *UBQLN2* which is associated with the formation of the inclusions, *SQSTM1, TBK1, VCP*, and *C9orf72* protein, which interacts with the inclusions previously formed [131,170,171,172].

Another mechanism that contributes to the pathogenesis of ALS is abnormal RNA metabolism, which includes abnormalities in RNA processing and miRNA expression that lead to misfolding proteins and formation of aggregates. RNA processing includes many mechanisms, such as transcription, splicing, editing, transport, and degradation. The key elements implicated in the deviations are RNA-binding proteins such as *TDP-43* and *FUS*, but there are additional targeted proteins that can present mutations, such as *ANG, STX, ATXN2, MATR3, hnRNPA1*, and *hnRNPA2B1*. This demonstrates that any mutations occurring at this level have an important role in ALS, although we cannot state that these mutations lead to the apparition of ALS [173].

Specific gene mutations that may contribute to ALS are *PFN1, SOD1*, and *TUBA4A*. The mutations of the C-terminus of kinesin-1 may also contribute, with the implicated mechanism affecting the integrity of the cytoarchitecture and the transport throughout the axon by destabilizing the tubulin networks, which may also affect the molecules carried inside the axon. The binding of the molecules transported inside are stabilized by *DCTN1*. It is mentioned that mutations at this level are not very common and do not play an important role in familial ALS apparition, but they rather contribute to diagnosing several ND diseases, especially in late-onset Parkinson’s, ALS, and FTD [131,174,175].

The different motor phenotypes of ALS can be classified based on whether upper or lower motor neurons are affected, and which regions they impact. Each subtype may present with differing life expectancies; some may even present with cognitive and behavioral deficits. Classic ALS is the most prevalent subtype, marked by signs of both upper and lower motor neuron loss in various body areas. By contrast, primary lateral sclerosis (PLS) typically presents as progressive spasticity with slowing movements, as well as isolated upper motor neuron signs. PLS patients exhibit no muscle atrophy, visible fasciculations, or denervation on EMG four years post-symptom onset. Their median survival is over 20 years, while PLS can transition into ALS three to four years after disease onset. A subtype called UMN-predominant ALS shows some LMN involvement, but less prominently than classic ALS. Patients in this subtype progress more slowly but live shorter lives than PLS patients [176,177].

There are multiple motor phenotypes of ALS, defined by their degree of involvement with upper versus lower motor neurons and regional distribution of symptoms. Understanding each subtype’s life expectancy and degree of cognitive and behavioral impairment is imperative.

One form of ALS, called LMN predominant ALS, involves limited UMN involvement with variable rates of progression. PMA (progressive muscular atrophy), on the other hand, involves progressive isolated LMN signs without evidence of UMN dysfunction. Up to 30% of PMA patients may develop these signs during follow-up.

Bulbar ALS is a devastating form of the disease characterized by rapid decline and an average lifespan of only two years from disease onset. It manifests as spastic dysarthria due to bulbar UMN dysfunction and tongue wasting and fasciculation due to bulbar LMN dysfunction. While only 30% initially exhibit bulbar symptoms, most ultimately experience difficulty speaking or swallowing due to this form of the condition.

Pseudobulbar Palsy (PBP) is another subtype, distinguished by absent facial expressions, spastic dysarthria, difficulty chewing, dysphagia, and tongue protrusion due to spasticity, but no fasciculation or wasting. This disorder originates in the upper motor neurons (UMNs), distinguishing it from progressive bulbar Palsy, which only affects LMNs but may not be universally recognized [178,179].

### 5.1. Perspectives for Treatment

The complexity of ALS has been demonstrated by its failure in over 40 randomized controlled trials that attempted to find effective disease-modifying medications. Riluzole is currently approved in most European countries as the sole disease-modifying medication; administered twice daily at 50 mg dose, its antiglutamatergic properties increase mean patient survival by approximately six months. The most frequently experienced side effects include nausea, diarrhea, fatigue dizziness, and liver issues.

More recently, edaravone (a free radical scavenger) has been studied in ALS patients. A phase III, double-blind study administering 60 mg/day of edaravone intravenously over 2 weeks per month resulted in significantly less decline in scores on the revised ALS Functional Rating Scale (ALSFRS-R). After six months, significantly fewer scores had decreased [180,181,182].

Masitinib, an oral tyrosine kinase inhibitor, is currently under study as a possible treatment for ALS. A randomized controlled trial administered masitinib as an add-on therapy with riluzole at 4.5 mg/kg/day, and showed positive effects in decreasing ALSFRS-R scores, especially among those experiencing typical disease progression. Its effectiveness will be further examined in a confirmatory study [183].

Honokiol was found to provide neuroprotective benefits by mitigating oxidative stress and improving mitochondrial function in affected neurons. Researchers also observed increased expression of key antioxidant enzymes and reduced expression of pro-inflammatory cytokines in those neurons treated with honokiol.

Overall, studies suggest that honokiol could serve as an effective therapy to treat ALS by decreasing oxidative stress and improving mitochondrial function in affected neurons. More research should be conducted in order to confirm these results, as well as to determine optimal dosing and treatment regimens for honokiol in ALS patients [147].

### 5.2. Brief Reflection Point

This section highlights the fact that ALS is a multifactorial disorder affected by genetic, environmental, and lifestyle influences; thus, integrating genetic/molecular knowledge with larger biopsychosocial models is vital in order to gain an overall understanding of this illness. At its core, ALS research is about helping those living with this devastating condition improve their quality of life, and eventually finding a cure. While great strides have been taken in understanding it thus far, much more work remains. Ongoing multidisciplinary research efforts must continue; each step forward in research contributes toward making patients’ lives better while ultimately finding an endpoint cure for this devastating illness.

## 6. Conclusions and Future Directions

Over the last five years, tremendous advances have been made in understanding both the pathophysiology and genetic basis of AD. Revamp of the amyloid-β cascade hypothesis and greater insight into AD preclinical phases has enabled improved comprehension. Genetic investigations have progressed from identifying three causal and one risk gene, to discovering multiple genetic markers that can be used to create a polygenic risk score for AD. Biomarker diagnosis has fundamentally transformed how AD is classified, making it possible to enroll patients earlier in studies when blood biomarkers become accessible. Molecular imaging will enhance diagnostic categorization and pathophysiology of any disease by providing visual evidence of co-pathology or regional protein aggregated deposits. At this rate, insights into risk reduction, primary and secondary prevention, and non-pharmacological and pharmacological treatments could become available and in parallel earlier than ever before. The early identification and multimodal treatment of patients could soon become a reality [184].

PD has been recognized for over 200 years and presents serious healthcare challenges worldwide. However, Parkinson’s is treatable when interventions are tailored specifically for each patient by trained healthcare providers. Here we highlight several promising advancements in Parkinson’s research and treatments which offer hope that services will continue to evolve to make a meaningful difference to those living with the disorder worldwide [185].

Understanding of Huntington’s disease (HD) has advanced substantially due to advances in genetic technology and large cohorts of individuals living with HD. Thus, new genetic modifiers of the disease have been identified. Somatic instability of CAG repeats is most prevalent in tissues most susceptible to HD pathology, and its degree is negatively correlated with age at disease onset. DNA repair components involved with mismatch repair may act to control somatic instability and the disease course. MutSb’s attempts at repair may induce loop-outs within CAG tracts targeted by MutSb, leading to potentially significant expansion. Reducing *MSH3, PMS2*, and *LIG1* pro-instability factors or hindering their function is thought to reduce somatic instability while being well-tolerated. *FAN1* expression decreases somatic instability while postponing disease onset; its upregulation could serve as protection from HD. Modulating these DNA repair components may also reduce instability in other pathogenic repeat sequences, suggesting these potential therapeutic options could also prove effective against repeat expansion diseases. Additionally, *mHTT* sequesters components of the NPC in aggregates, disrupting nucleocytoplasmic transport. Modulating nuclear transport pathways has proven protective in cell models of HD, opening new avenues for therapeutic intervention. Cerebrospinal fluid (CSF) offers an ideal source of CNS proteins for clinical trials due to its accessibility throughout. Notably, neurofilament light chain (NfL) is one of the proteins released into CSF and plasma after neuronal damage to be used as a biomarker or surrogate endpoint for clinical trials. Its levels seem to correlate strongly with disease progression; thus, its levels could serve both functions simultaneously. Furthermore, *mHTT* could also be released by damaged neurons, with CSF concentration changes representing early changes to be found prior to manifestation of HD [112,186].

In recent years we have witnessed an increased focus on developing precision medicine approaches for ALS subtypes with known genetic causes. One promising therapeutic avenue involves antisense oligonucleotides (ASOs), which are short sequences of nucleotides that can modulate gene expression and splicing. ASOs have proven effective against preclinical models of ALS characterized by *SOD1* mutations and *C9orf72* repeat expansions; clinical studies utilizing intrathecal administration of ASOs targeting these genes are currently ongoing. Stem cell therapies such as granulocyte-colony stimulating factor (G-CSF) are being explored as possible therapies for ALS. Peripheral blood stem cells, bone marrow mesenchymal stem cells, and non-neural progenitor cells induced by granulocyte-colony stimulating factor have all shown safety and tolerability when administered to ALS patients. However, their efficacy against disease progression remains undetermined. Current phase II and III clinical trials offer hope that more precise classification of ALS cases based on pathogenic mechanisms will lead to targeted therapies with favorable results in particular ALS subgroups, eventually making ALS disease manageable [187,188].

## Figures and Tables

**Figure 1 ijms-24-10809-f001:**
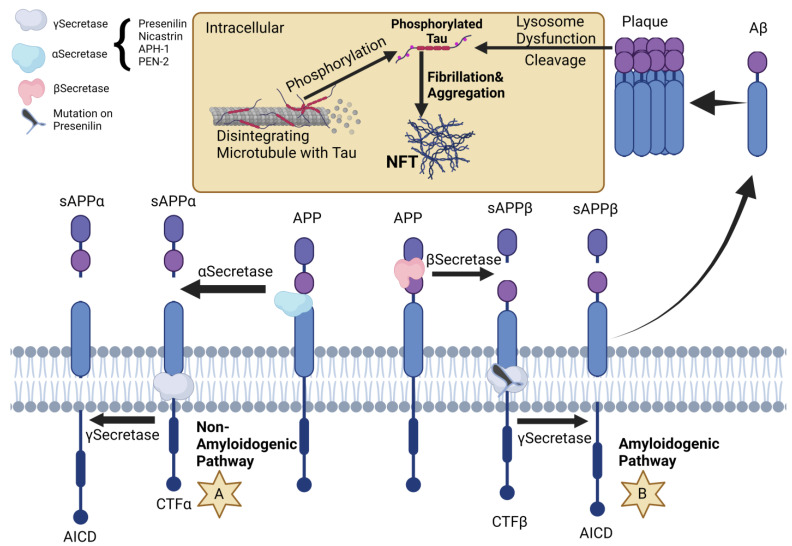
Amyloid precursor protein (APP) is a protein that spans the cell membrane. Its processing can occur through two pathways: the non-amyloidogenic (**A**) and amyloidogenic (**B**) pathways. In the non-amyloidogenic pathway, APP is cleaved in the middle of Aβ by α-secretase, resulting in the production of soluble APPα (sAPPα) and C-terminal fragment α (CTFα), which is then hydrolyzed by γ-secretase to generate the APP intracellular domain (AICD). In the amyloidogenic pathway, APP is cleaved by β-secretase, resulting in the release of N-terminal soluble APPβ (sAPPβ) and the C-terminal fragment β (CTFβ), which is then hydrolyzed by γ-secretase to produce Aβ and AICD. γ-secretase is composed of several parts, including presenilin, nicastrin, anterior pharynx-defective 1 (APH-1), and presenilin enhancer 2 (*PEN-2*). Mutations in the PSEN gene may increase the activity of γ-secretase, leading to the formation of plaques. Moreover, the plaques lead to lysosomal dysfunction (interacting with Caspaze 2/3), promoting cleavage of Tau and NFT (neurofibrillary tangles) formation. Created with BioRender.com.

**Figure 2 ijms-24-10809-f002:**
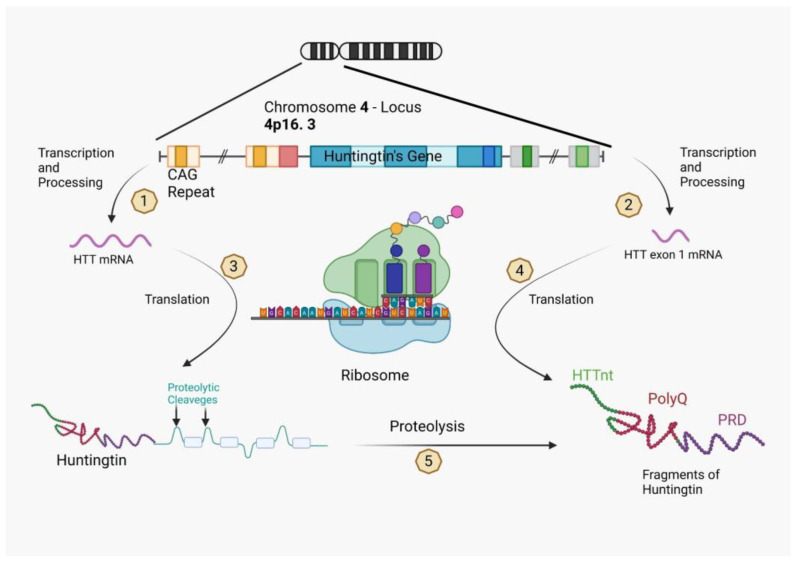
The huntingtin protein can take on various forms and undergo changes depending on the expression of the HTT gene. Normally, the expression of HTT results in the production of an RNA transcript that codes for the complete huntingtin protein. However, if the gene has an expanded CAG repeat, the RNA transcript can be processed abnormally, producing an mRNA that encodes only the HTT exon1 protein. 1 & 2—Transcription and Processing; 3 & 4—Translation; 5—Proteolysis. Created with BioRender.com.

**Figure 3 ijms-24-10809-f003:**
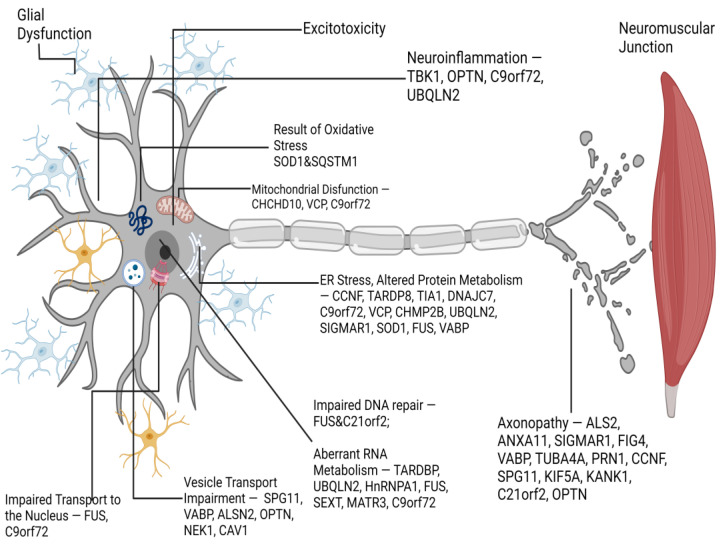
Advances in large-scale genomic analysis have uncovered a variety of causative genes and risk factors for amyotrophic lateral sclerosis (ALS). These gene variants map onto key pathogenic mechanisms relevant to all motor neuron cellular compartments as well as neighboring cells such as glia and interneurons. In this way, these mechanisms are genetically validated, enabling a greater confidence in their targeting for therapeutic benefit. Some of these mechanisms have emerged only in recent years due to new genetic information, including gene changes highlighting dysregulation of RNA processing and metabolism. There is significant overlap of some genes with those found in closely related disorders such as frontotemporal dementia (for example, *C9orf72*, *CHCHD10*, *SQSTM1*, *TBK1*, *CCNF*, *FUS*, *TARDBP*, *OPTN*, *UBQLN2*, *TUBA4A*, *ATAXN2*, *VCP*, and *CHMP2B*). This suggests a closer relationship with broader neurodegenerative disorders, and indeed many of the pathways depicted are relevant in, for example, Alzheimer’s disease. ER, endoplasmic reticulum. Created with BioRender.com.

**Table 1 ijms-24-10809-t001:** Most prominent mutations in APP gene.

Mutation	Pathogenicity	Type of Mutation	Biological Effect	Citation
*A673T* (Icelandic)	Alzheimer’s Disease—Protective	Substitution	This particular type is linked to limited build-up of amyloid and is believed to guard against amyloid-related issues. It results in a decrease of approximately 40 percent in the production of amyloidogenic Aβ peptides, and the Aβ that is produced has a reduced tendency to form clumps.	[17,18,19,20]
*A673V*	Not Classified	Substitution	According to the CERAD criteria, a clear diagnosis of AD was made, as evidenced by substantial Aβ and tau pathology deposits (Braak stage VI) along with cerebral amyloid angiopathy. The deposits found contained elevated levels of Aβ40 and were notably larger, with fewer preamyloid deposits. Perivascular localization was frequently observed. In laboratory studies, it was discovered that A673V caused a shift in β-secretase processing of APP toward the amyloidogenic pathway and amplified Aβ aggregation.	[21,22]
*E693Q* (Dutch)	Hereditary Cerebral Hemorrhage with Amyloidosis—Pathogenic	Substitution	There is a substantial accumulation of amyloid in the cerebral blood vessels, accompanied by hemorrhages and some diffuse plaques in the brain tissue. In laboratory experiments, it was observed that this condition speeds up Aβ aggregation in vitro, leading to greater fibril formation, and may also modify APP processing.	[24,25]
*E693del* (Osaka, E693∆, *E693delta*	Alzheimer’s Disease—Pathogenic	Deletion	This variant led to an increased oligomerization and nucleation of Aβ aggregates in vitro. Furthermore, it was found that there was no alteration in the Aβ42/Aβ40 ratio, but there was a decrease in both Aβ42 and Aβ40. This variant was also discovered to be more resistant to degradation by neprilysin and insulin-degrading enzyme. Additionally, this variant had a greater inhibitory effect on long-term potentiation (LTP) compared to wild-type Aβ, which suggests a potential negative impact on synaptic plasticity.	[26,27,28,29]
*E693K* (Italian)	Hereditary Cerebral Hemorrhage with Amyloidosis—Pathogenic	Substitution	The observed symptoms include small to large hematomas, subarachnoid bleeding, scars with hemosiderin deposits, small infarcts, and cortical calcifications. Aβ immunoreactivity was observed in vessel walls and neuropil, but there was an absence of neurofibrillary changes and neuritic plaques. Despite a reduction in the Aβ42/Aβ40 ratio and a decrease in Aβ42 levels, the mutant peptide was found to be toxic in cells and aggregates at a faster rate.	[23]
*E693G* (Arctic, E22G)	Alzheimer’s Disease—Pathogenic	Substitution	Several carriers displayed neuropathology that was indicative of AD. Plaques were observed to have a “targetoid” shape, containing heterogeneous truncated Aβ peptides in the center and surrounded by Aβ42. Cell-based assays revealed a reduction in the production of both Aβ40 and Aβ42. Additionally, there was a decrease in proteolytic degradation of Aβ by neprilysin, a type of enzyme that breaks down proteins.	[23,25]
*c.-488C>A* (rs532314089)	Alzheimer’s Disease	Substitution	Predicted to disrupt binding of transcription factor EGR1. PHRED-scaled CADD = 0.26. Negative regulator in multiple cell types including PC12 neuronal-like rat chromaffin cells, SK-N-SH neuroblastoma cells, C6 glial cells and U373 astroctyoma cells among others	[30,31,32]
*c.24+38G>A* (rs373985746)	Alzheimer’s Disease	Substitution	Predicted benign in silico (PHRED-scaled CADD = 10).	[32]
*c.24+288G>A* (rs192348494)	Alzheimer’s Disease	Substitution	Predicted benign in silico (PHRED-scaled CADD = 12).	[32]
*c.-23-377A>G* (rs150375400)	Alzheimer’s Disease	Substitution	Predicted benign in silico (PHRED-scaled CADD = 10).	[32]
*A18T*	Alzheimer’s Disease, Cardiovascular Disease	Substitution	Predicted to disrupt signal peptide cleavage and affect APOE secretion. PHRED-scaled CADD = 22.	[32,33]

CADD—Combined Annotation-Dependent Depletion.

**Table 2 ijms-24-10809-t002:** Recent prominent mutations in the *PSEN2* gene.

Mutation	Pathogenicity	Type of Mutation	Biological Effect	Citation
*A79V*	Alzheimer’s Disease —Pathogenic	Substitution	The observed neuropathology was in line with that of AD. It was observed that this variant led to an increase in the Aβ42/Aβ40 ratio and a decrease in the Aβ37/Aβ42 ratios in cells.	[39,40,41]
*M84V*	Alzheimer’s Disease —Pathogenic	Substitution	In two cases, the observed neuropathology was consistent with AD. Additionally, MRI scans revealed cortical and cerebellar atrophy in these two cases. In the third case, frontal and temporal lobe atrophy was observed. Cell studies revealed an increase in both Aβ42 and the Aβ42/Aβ40 ratio.	[42,43]
*L85P*	Alzheimer’s Disease —Pathogenic	Substitution	SPECT and PET scans showed bilateral hypoperfusion and hypometabolism in the occipital and temporal lobes. Cell studies revealed an increase in the Aβ42/Aβ40 ratio as well as increased Aβ42 levels in transfected cells. In vitro studies indicated a decrease in Aβ42 production and the complete absence of Aβ40 production.	[44,45]
*L113_I114insT* (int4del)	Alzheimer’s Disease —Pathogenic	Substitution	The observed neuropathology was consistent with AD, and included neuron loss in the hippocampus and entorhinal cortex, the presence of neuritic plaques and neurofibrillary tangles in the hippocampus, and amyloid angiopathy, which was particularly evident in the cerebellum. The identified mutation involved a deletion of a G in the splice donor site of intron 4, resulting in the production of three aberrant transcripts. Further investigations indicated an increase in both Aβ42 and the Aβ42/Aβ40 ratio, as well as a reduction in Aβ40 and Aβ38 production in patient brain membranes.	[46,47,48]
*M139V*	Alzheimer’s Disease —Pathogenic	Substitution	Decrease in the levels of Aβ40, Aβ38, and Aβ37, and an increase in the levels of Aβ42 and Aβ43. In iPSC-derived neurons, the levels of mutant protein were found to be variable, suggesting protein instability.	[49,50,51]

**Table 3 ijms-24-10809-t003:** Recently discovered new APOE gene mutations.

Mutation	Pathogenicity	Type of Mutation	Biological Effect	Citation
*K82fs*	Tauopathy and Pick’s Disease	Deletion	The neuropathological findings were consistent with Pick’s disease. A frameshift was identified to start at K82, and the mutant protein was found to be reduced in the frontal cortex and hippocampus.	[54]
*c.*71C>A*	Alzheimer’s Disease —Pathogenic	Substitution	In one case, an MRI scan revealed widening of the sulcus, fissure, and temporal horn, along with a decrease in hippocampal volume. Additionally, FDG-PET showed hypometabolism in the bilateral frontal, parietal, and temporal lobes. Among the five affected carriers, CSF analysis showed Aβ42, total tau, and phospho-tau levels consistent with AD. The study suggests a possible reduction in the binding of PSEN2 expression suppressor miR-183-5p, which may lead to an increased Aβ42/Aβ40 ratio.	[55,56]
*M239V*	Alzheimer’s Disease —Pathogenic	Substitution	The brain pathology showed diffuse cerebral atrophy, senile plaques, neurofibrillary tangles (Braak and Braak stage VI), ectopic neurons in subcortical white matter, and extracellular “ghost” neurofibrillary tangles. In cell-based assays, there was an increase in the Aβ42/Aβ40 ratio and an increase in Aβ42 levels. However, there was no change in the proteolytic products PSEN2-CTF and PSEN2-NTF.	[57,58]

**Table 4 ijms-24-10809-t004:** Most relevant mutations in the MAPT gene leading to AD.

Mutation	Pathogenicity	Type of Mutation	Biological Effect	Citation
*IVS10+12 C>T*	Familial Danish Dementia—Pathogenic	Substitution	The mutant protein leads to the formation of tau aggregates in both neurons and glia, and isolated tau filaments exhibit a twisted, ribbon-like morphology and consist of hyperphosphorylated 4-repeat (4R) tau isoforms. The mutation also causes a destabilization of a stem-loop structure that regulates the alternative splicing of exon 10, resulting in a higher frequency of inclusion of exon 10 and an increased proportion of 4R tau isoforms.	[84,85]
*A152T*	Alzheimer’s Disease—Risk	Substitution	The presence of tau pathology is a common feature, often accompanied by Lewy bodies, amyloid plaques, or TDP-43 pathology. The mutant tau has a decreased ability to bind to microtubules, leading to less efficient microtubule assembly and impaired microtubule stability. Additionally, it has an increased propensity to form tau oligomers and is more susceptible to proteolysis by caspases.	[86,87,88]
*K257T*	Tauopathy and Frontotemporal —Pathogenic	Substitution	The patient exhibited frontotemporal atrophy with significant temporal lobe involvement. Tau-positive Pick bodies were found in the neocortex, hippocampus, and subcortical regions similar to those seen in sporadic Pick’s disease. Some cell bodies showed diffuse hyperphosphorylated tau. In vitro analysis showed that recombinant tau protein with the K257T mutation had a decreased ability to promote microtubule assembly.	[89]
*L266V*	Frontotemporal —Pathogenic	Substitution	The patient had severe atrophy of the frontal and temporal lobes, with extensive neuronal loss and gliosis. Tau-positive inclusions, including Pick bodies, and tau-positive argyrophilic astrocytes with stout filaments and round or irregular argyrophilic inclusions were also observed. In molecular studies, there were increased levels of exon 10+ tau mRNA and soluble four-repeat (4R) tau. The patient showed a decreased rate and extent of tau-induced microtubule assembly, as well as a specific increase in tau self-assembly for the 3R isoform.	[90,91]

**Table 5 ijms-24-10809-t005:** Important SOD1 mutations associated with ALS onset.

Mutation	Pathogenicity	Type of Mutation	Biological Effect	Citation
*A4V*	ALS Pathogenic	Substitution	This mutation is responsible for a rapidly progressive dominant form of amyotrophic lateral sclerosis (ALS) that exclusively affects lower motor neurons, and it accounts for 50% of SOD1 mutations associated with familial ALS in North America. However, it is a rare mutation in Europe.	[142,146]
*G93A*	ALS Pathogenic	Substitution	Patients showing different oxidative markers, such as glutamate excitotoxicity, and dysfunctions at several levels, such as mitochondria, due to calcium influx, and axon as well as protein oxidation; modifications were observed at SOD1-G93A in mice. This mutation is relatively rare in the general population but it is very common in familial ALS, and multiple studies on animal models have also shown that having the SOD1-G93A mutation is enough to cause motor-neuron degeneration.	[147,148,149]
*L84F*	ALS Pathogenic	Substitution	Protein instability and misfolding that can lead to forming protein accumulations.	[144,145]

**Table 6 ijms-24-10809-t006:** Important TARDBP mutations associated with ALS onset.

Mutation	Pathogenicity	Type of Mutation	Biological Effect	Citation
*M337V*	ALS Pathogenic	Substitution	Production of an abnormal TDP-43 protein that aggregates abnormally and accumulates in motor neurons, ultimately leading to their degeneration and death, and thus contributing to ALS development.	[150,154,155]
*A315T*	ALS Pathogenic	Substitution	Transgenic mice carrying the A315T mutation of TDP-43 may succumb to early death due to digestive complications before fully manifesting neurological signs associated with ALS, suggesting it also influences their digestive systems and may contribute to their early demise. Although the exact mechanisms underlying gastrointestinal complications remain poorly understood, experts speculate that abnormal TDP-43 protein build-up in intestinal cells may lead to dysfunction and damage.	[156]
*A382T*	Possible ALS Pathogenic	Substitution	Unknown mechanism.	[157,158]

## Data Availability

All data is available on PubMed and on ALZFORUM database.

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
