# Peer review of "Unraveling Molecular and Genetic Insights into Neurodegenerative Diseases: Advances in Understanding Alzheimer’s, Parkinson’s, and Huntington’s Diseases and Amyotrophic Lateral Sclerosis"

_ijms, 2023, doi:10.3390/ijms241310809_

Round 1

Reviewer 1 Report

Interesting concept to provide up-to-date information on neurodegenerative diseases. However, despite its lengthy chapter-like compilation, I have some hesitation in approving this version for publication.

Firstly, the review appears to be a compilation of previously existing reviews in the separate field of neurodegenerative diseases. It is filled with typographical errors, lacks clarity, conveys information superficially, and contains some technical errors. At this point, I would not recommend this review. It needs to be thoroughly revised for further evaluation.

Include studies with non coding RNA in the filed

Nagaraj S, Zoltowska KM, Laskowska-Kaszub K, Wojda U. microRNA diagnostic panel for Alzheimer’s disease and epigenetic trade-off between neurodegeneration and cancer. Ageing research reviews. 2019 Jan 1;49:125-43.

Takousis P, Sadlon A, Schulz J, Wohlers I, Dobricic V, Middleton L, Lill CM, Perneczky R, Bertram L. Differential expression of microRNAs in Alzheimer's disease brain, blood, and cerebrospinal fluid. Alzheimer's & Dementia. 2019 Nov 1;15(11):1468-77.

Swarbrick S, Wragg N, Ghosh S, Stolzing A. Systematic review of miRNA as biomarkers in Alzheimer’s disease. Molecular neurobiology. 2019 Sep 15;56:6156-67.

Cortini F, Roma F, Villa C. Emerging roles of long non-coding RNAs in the pathogenesis of Alzheimer’s disease. Ageing research reviews. 2019 Mar 1;50:19-26.

Several relevant citations are missing on multiple occasions. It is hard to refer back the relevant studies. 

Table 2 is labeled as PSEN1, but it actually pertains to APP.

Figure 1 does not provide any information on the Tau pathology.

It is recommended to improve the English throughout the manuscript.

In Figure 1, please indicate the loci on the chromosome and extend the representation. The entire chromosome is not related to HTT. The current diagram lacks meaningful information.

It is recommended to create a single Figure 1 that includes all the genes and their mutations.

"NFT" refers to neurofibrillary tangles, not "neural focal thresholds." It is unclear if the authors are aware of the pathology mechanism.

Typographical errors can be found on lines 649, 698, 755, 792, and 788.

MiR-183-5P is not an inhibitor of PSEN2; it represses PSEN2 expression through base pairing.

The other errors are:

Line 44: "...what Charles Darwing later..."

Line 86: "...which finished with..."

References missing on lines 106 and 134.

Inconsistent use of APOE, apoe, and Apoe.

Inconsistent convention for dosage: 10 mg/kg or 10mg/kg.

Repetition of the same text from line 359 to 388.

Missing reference for Vincent and Davies paper in lines 454 to 463 (similar issues occur in other instances).

No reference provided in lines 571 to 578.

Only Donanemab is discussed, but the title implies a more general topic.

In line 898, specify the test conducted for post hoc analysis.

Lines 938 to 940 contain incorrect and vague information. Please rephrase appropriately.

In line 1006, is "ARN" referring to RNA?

Correct the reference styling in lines 1019 and 1023.

Use the same convention for amino acids in lines 1158 and 1161.

No reference is provided in lines 1173 to 1176. What does "This article" refer to?

Only a few errors have been mentioned. However, it is strongly advised that the authors carefully review the manuscript for English language proficiency, correct any typos, and ensure scientific accuracy before resubmitting it. Paying close attention to these aspects will greatly enhance the quality of the manuscript.

Only a few errors have been mentioned. However, it is strongly advised that the authors carefully review the manuscript for English language proficiency, correct any typos, and ensure scientific accuracy before resubmitting it. Paying close attention to these aspects will greatly enhance the quality of the manuscript.

Reviewer 2 Report

The review of  Ciurea  and co-workers is very interesting. 

Minor revisions: 

Mistakes in the number of paragraph. Presenilin should be 4 and also for the others.

Probably authors should reorganized the paragraph as: 2. Alzheimer and 2.1; 2.2: 2.3  and so on for the genetic variations that correlates with the  diseases. 

I added some minor comments  in the text

Reviewer 3 Report

The authors summarize the evidence for the molecular and genetic Insights into the most common neurodegenerative diseases, including Alzheimer's, Parkinson's, Amyotrophic Lateral Sclerosis, and Huntington's Disease. They list prior studies on the molecular and genetic background of the related pathogenesis events and therapy perspectives. The authors provided three interesting figures that summarize the amyloid precursor protein (APP), huntingtin protein, and gene variants in the neuropathology of amyotrophic lateral sclerosis. Moreover, the authors provided seven interesting tables that summarize the most prominent mutations in APP gene, prominent mutations in PSEN1 gene, etc. From this discussion, the authors suggest that significant advances have been made in understanding both the neuropathology and genetic basis of the four degenerative diseases. The authors also concluded that novel therapies that target specific genes could be useful for future therapy. The summarized information is interesting, and the review is well-written.

Comments:       

1) The abbreviation” ALS” in the title needs to be written as a full name.

2) The provided sections read like narration for the evidence of discussed points without critical aspects/reflection points. At the end of each section, a take-home message is advised to be provided.

3) To avoid confusion of readers, the authors are advised to clearly describe in the narration of previous literature whether these data are derived from clinical studies or from experimental studies. This point needs to be carefully addressed by the authors in the entire manuscript.

4) The work lacks future directions that will include limitations and what is the next step to translate these findings to clinical settings.

5) In Figure 1, the authors are advised to make a visual distinction between the amyloidogenic vs. non-amyloidogenic pathway by marking the “non-amyloidogenic pathway” with A and the “amyloidogenic pathway” with B. Also, add these letters to the figure caption.

6) In order to attract the interest of more readers regarding the current review, the authors are advised to summarize the narration with additional figures, in the same way, they did for Figures 1-3.  

7) The authors are advised to make the table/figure captions stand-alone. To this end, authors are advised to provide a list of abbreviations describing the full names of all the listed abbreviations in the table/figure.

8) In line 1052, please add the caption title to Table 6.

9) Likewise, in line 1120, please add the caption title to Table 7.

10) The numbering of the manuscript section is confusing. Numbering using subsection numbers is recommended. The manuscript should be re-numbered as 1. Introduction, 2. Alzheimer’s disease, 2.1. Amyloid Precursor Protein, 2.2., 2.3., … 3. Parkinson’s disease (PD), 4., 5., etc.  

11) In line 1279, the “discussions” sections should be replaced by “Conclusions and future directions”.

12) In line 106, the authors state “a study conducted in 2012 by Jonsson, T., Atwal, J.,Steinberg, S. et al.”. The authors are advised to re-write the statement as “a study conducted in 2012 by Jonsson et al. [citation number]”.

13) In line 549, the citation number should be added immediately after “Lowe et al.”.

14) The citation style within the text is wrong and needs to be reformatted according to the instructions of the journal.

15) Careful revision of the reference list should be performed. For example, reference no. 8 includes the authors’ affiliations.  

16) Some typos/syntax errors are present in the manuscript which need to be addressed, for example:

- In line 149, the legend title for table 1 reads as “Table 1. Most proeminent mutations in APP gene”. Please, correct “proeminent” to “prominent”. Kindly, address this point in the entire manuscript.

Moderate editing of the English language is required.

Round 2

Reviewer 1 Report

The current version is improved but still few comments  need to be addressed

To enhance clarity, consider unifying the labeling of Table 1 and Table 2, as they currently have the same label.

Figure 1 lacks information on Tau pathology. To address this, draw representations of hyperphosphorylated proteins and NFT (neurofibrillary tangles) to indicate tau pathology. Include visually informative elements such as labeled structures or diagrams, illustrating the presence of hyperphosphorylated Tau proteins and their aggregation into NFTs. By incorporating this information, Figure 1 will provide a more comprehensive overview of the Tau pathology.

It is recommended to create a single figure, encompassing all the genes (APP, PSEN, APOE4, and MAPT), along with their respective mutations in the amino acid sequences
